# Geochemical studies on rock varnish and petroglyphs in the Owens and Rose Valleys, California

**Meinrat O. Andreae**[1,2,3]*, **Abdullah Al-Amri**[2], **Tracey W. Andreae**[1], **Alan Garfinkel**[4], **Gerald Haug**[1], **Klaus Peter Jochum**[1], **Brigitte Stoll**[1], **Ulrike Weis**[1]

1 Max Planck Institute for Chemistry, Mainz, Germany, 2 Department of Geology and Geophysics, King Saud University, Riyadh, Saudi Arabia, 3 Scripps Institution of Oceanography, UCSD, La Jolla, California, United States of America, 4 California Rock Art Foundation, Bakersfield, California, United States of America

* m.andreae@mpic.de

**Data Availability Statement:** All data files are available from the Edmond database (https://edmond.mpdl.mpg.de/imeji/collection/JO7ClPMzy4AkczTr).

## Abstract

We investigated rock varnish, a thin, manganese- and iron-rich, dark surface crust, on basaltic lava flows and petroglyphs in the Owens and Rose Valleys (California) by portable X-ray fluorescence (pXRF) and femtosecond laser-ablation inductively-coupled-plasma mass spectrometry (fs-LA-ICPMS). The major element composition of the varnish was consistent with a mixture of Mn-Fe oxyhydroxides and clay minerals. As expected, it contained elevated concentrations of elements that are typically enriched in rock varnish, e.g., Mn, Pb, Ba, Ce, and Co, but also showed unusually high enrichments in U, Cu, and Th. The rare earth and yttrium (REY) enrichment pattern revealed a very strong positive cerium (Ce) anomaly and distinct negative europium (Eu) and Y anomalies. The light rare earth elements (REE) were much more strongly enriched than the heavy REY. These enrichment patterns are consistent with a formation mechanism by leaching of Mn and trace elements from aeolian dust, reprecipitation of Mn and Fe as oxyhydroxides, and scavenging of trace elements by these oxyhydroxides. We inferred accumulation rates of Mn and Fe in the varnish from their areal densities measured by pXRF and the known ages of some of the lava flow surfaces. The areal densities of Mn and Fe, as well as their accumulation rates, were comparable to our previous results from the desert of Saudi Arabia. There was a moderate dependence of the Mn areal density on the inclination of the rock surfaces, but no relationship to its cardinal orientation. We attempted to use the degree of varnish regrowth on the rock art surfaces as an estimate of their age. While an absolute dating of the petroglyphs was not possible because of the lack of suitable calibration surfaces and a considerable amount of variability, the measured degree of varnish regrowth on the various petroglyphs was consistent with chronologies based on archeological and other archaeometric techniques. In particular, our results suggest that rock art creation in the study area continued over an extended period of time, possibly starting around the Pleistocene/Holocene transition and extending into the last few centuries.

**Funding:** MOA received support from the German Max Planck Society (https://www.mpg.de/en) and from King Saud University (https://ksu.edu.sa/en/). No specific grant numbers were assigned. The funders had no role in study design, data collection and analysis, decision to publish, or preparation of the manuscript.

**Competing interests:** The authors have declared that no competing interests exist.

## Introduction

Rock varnish, first scientifically described by Alexander von Humboldt from the Orinoco River in Venezuela [1], is a thin, dark, manganese-rich coating that is found in a variety of environments across the earth and even on Mars [2–6, 7; and references therein]. It consists of a matrix of poorly crystallized manganese (Mn) and iron (Fe) oxides and hydroxides (oxyhydroxides), in which clay and other detrital minerals are embedded [3, 8–11]. In a previous publication [7], we have proposed a classification of rock varnish into five types based on its growth environment, microstructure, and chemical composition, ranging from "desert varnish" (Type I), commonly found in arid regions, to the "river varnish" (Type V) that forms on rocks in the splash zones of many rivers and represents the varnish described by von Humboldt.

This paper focuses on Type I varnish, which is generally characterized by a layered structure, high rare earth element (REE) and barium (Ba) enrichments, and birnessite as the dominant Mn mineral [7]. The details of the processes by which rock varnish is formed are still the subject of ongoing scientific debate, but there is a developing consensus that the Mn and other enriched elements in the varnish matrix, as well as the embedded detrital minerals, are derived from dust deposition and are transformed into the varnish coating by a sequence of dissolution and re-precipitation events, which may involve abiotic reactions, microbial catalysis, and/or photo-oxidation [e.g., 12, 13–16]. Further detail and references on this issue can be found in our previous papers [7, 17, 18].

In arid environments, varnish is found on rock surfaces that have been exposed over a wide range of ages, from centuries to hundreds of thousands of years. Its thickness is typically in the range of tens to a hundred μm, and even on the oldest surfaces it rarely exceeds 200 μm [19]. Somewhat surprisingly, the varnish thicknesses in Liu and Broecker's [19] data set do not show a general relationship with age, with varnishes that are only ~10,000 years old being just as thick as those ten to twenty times older. In part, this may be due to Liu and Broecker's sampling strategy, as they were measuring the varnish thickness in the deepest part of microbasins, millimeter-sized depressions in the rock surface where varnish has accumulated relatively undisturbed. It may also reflect the possibility that once a certain varnish thickness has been reached, loss by processes like cracking and spalling prevents further thickness growth [20–22]. This is supported by the fact that the oldest varnishes in Liu and Broecker's [19] data set (>50 ka) show the lowest apparent growth rates (0.6–2.0 μm ka$^{-1}$), whereas the Holocene varnishes all appear to have much higher rates (12–40 μm ka$^{-1}$).

Our studies on the growth of rock varnish have been motivated in part by an interest in using the growth rate of varnish on rock surfaces to obtain age estimates for rock art (petroglyphs) and inscriptions. Petroglyphs are produced by removing the dark varnish by abrasion, scratching, or pecking, thereby exposing the lighter surface of the underlying rock. Such images are found worldwide and have been created from the pre-Neolithic period up to today [e.g., 23, 24–31]. Once the petroglyph has been created, varnish begins to deposit again on the exposed fresh rock surface and, if the rate of accumulation were known, the amount of the regrown varnish could be used to date the rock art. Such a technique would be highly desirable, given the archeological importance of rock art as an expression of ancient cultures and the difficulties encountered in attempts to date petroglyphs by other means [summarized in 17, and refernces therein].

Using varnish regrowth as an indicator of age has been applied frequently in a qualitative and relative way by visual comparison of the darkness of varnish on alluvial fans of different age or on superimposed petroglyphs [25, 32–36]. A quantitative version of this approach using colorimetric measurements was developed by Bednarik [32], who found "fairly good consistency" between color measurements and age. A similar technique was used to achieve a relative

chronological ordering of rock art elements at Little Lake, California, by Bretney [37]. The amount of Mn deposited on the rock surface or the Mn/Fe ratios in the varnish have been explored by other authors as potential age indicators [34, 38, 39]. However, all of these approaches have to be viewed with great caution, because the growth rate of varnish is highly variable and depends on a large number of parameters other than age, including the exposure of the rock surface to dust, erosion by wind and water, the orientation and slope of the rock surface, the hardness, roughness, and texture of the rock underneath, and its initial iron content [3, 19, 22, 34, 40, 41], as summarized in [17] and in S1 Table.

In two previous studies, we demonstrated the potential of measuring the amount of Mn in rock varnish on petroglyphs and adjacent intact rock surfaces by portable X-ray fluorescence (pXRF) as a tool to determine the growth rate of varnish and to estimate the age of rock art in Saudi Arabia [17, 42]. That work benefitted from the existence of distinct time markers in the Arabian rock art in the form of particular types of scripts, which had been used during specific time periods, and of dated paleoclimatic transitions that were reflected in the animal species depicted in the rock art [43–47]. Using these time markers, we found average Mn accumulation rates (i.e., the rate at which Mn accumulates in the form of varnish per unit area and time) of 17 and 13.4 ng cm$^{-2}$ a$^{-1}$ for our sites in the Ha'il (northwestern Arabia) and Hima (southwestern Arabia) regions, respectively, with confidence intervals of about a factor of two. We also derived a quantitative metric for the degree of varnish regrowth since the creation of the rock art by calculating the ratio of the Mn areal density (i.e., the mass of Mn in the varnish per unit area) within a petroglyph to the Mn density on the adjacent intact varnish, expressed as a percentage, which we refer to as the normalized Mn accumulation rate, $N_{Mn}$. We obtained $N_{Mn}$ values of 12±3 and 10.4±3% ka$^{-1}$ for the Ha'il and Hima regions, respectively, thus narrowing the variance significantly by this normalization approach. From these studies, we estimated the statistical uncertainty of an age estimate based on the $N_{Mn}$ measurements to be about 33%, but cautioned that numerous additional assumptions went into converting $N_{Mn}$ into an age estimate, and that for the time being, this approach must be considered experimental. The observed rates of regrowth of the varnish, on the order of 10% ka$^{-1}$, also imply that after about 10 ka, the areal density of the regrown varnish is indistinguishable from intact varnish, and therefore any potential dating application would be limited to the Holocene. We could show, however, that the age estimates so obtained were consistent with ages based on the cultural and ecological content of the rock art, and allowed a meaningful ordering of rock images into an age sequence.

Given the relative success of our approach in Saudi Arabia, we intended to examine the possibility of extending it to other regions. We chose the southern Owens Valley and the adjacent Rose Valley, both parts of the Mojave Desert in southern California, as an initial test site because of its prominent role in the scientific study of rock varnish [e.g., 2, 19], the existence of radiometrically dated rock surfaces [e.g., 19, 28, 48–50], and the presence of well-documented rock art [e.g., 24, 51], including a large corpus of dated petroglyphs in this and nearby regions [36, and references therein, 52]. The rock art in this area has been the subject of vigorous–and sometimes acerbic–debate over decades, both with regards to its meaning and function [e.g., 24, 51, 53–59] and its time of creation [e.g., 24, 28, 36, 52, 54, 55, 60, 61–63].

In this study, we conducted in-situ measurements by pXRF on lava flow surfaces of known age in the study region to determine the areal density, $D_{Mn}$ and $D_{Fe}$, of Mn and Fe on the surface of the rocks and to estimate the rate of accumulation of these elements in the form of rock varnish. We also examined the dependence of $D_{Mn}$ and $D_{Fe}$ on the cardinal orientation and slope of the rock surfaces. We analyzed selected varnish samples by femtosecond laser-ablation inductively-coupled-plasma mass-spectrometry (fs-LA-ICPMS) in order to characterize the geochemical signature of the varnish, determine its type according to the classification of

Macholdt et al. [7], and look for clues on its environment of formation. Finally, we measured the Mn and Fe areal densities on a number of petroglyphs to explore the potential for deriving age estimates.

## Material and methods

### Study region, climate, and history

Our study area is in the northwestern margin of the Mojave Desert, California, and includes sites in the southern part of the Owens Valley and in its southern continuation, Rose Valley. It lies NNW of Ridgecrest, CA, between the latitudes of 37˚N (Aberdeen part of the Big Pine volcanic field) and 36˚N (Little Lake and Fossil Falls sites), at a longitude of about 118˚W (overview map in S1 Fig). The region is in the rain shadow of the Sierra Nevada mountain range and has an arid to semiarid climate. Annual rainfall is 125 to 175 mm, mostly in the form of winter rains, and temperatures span a wide range, from -19 to 43˚C, with annual averages around 15–18˚C [64]. The dominant vegetation is dryland scrub and the soils are mostly relatively saline and alkaline. During most of the Pleistocene, the region was much wetter and cooler than today, with the transition to arid and semiarid conditions similar to the present climate taking place from about 12 to 6 ka BP (BP: before present, referring to the year 2000 CE), involving highly variable conditions [65–67]. An extensive drought period occurred between about 5000 and 4000 BP [68, and references therein]. In the late Holocene, megadroughts occurred in the Medieval Warm Period (MWP) around 1200–1350 CE, with subsequent wetter periods in the late MWP and the Little Ice Age around 1650 CE [66].

Human occupation in the region began around the Pleistocene-Holocene transition, ca. 13,000 BP, by a sparse population of hunter-gatherers utilizing mostly small mammals, dryland seeds, and marshland plants as their food source [69, 70; and references therein]. Seasonal migratory hunting-gathering prevailed as the dominant economic pattern in the region into the ethnographic period, supplemented by some small-scale agriculture beginning around 2000 BP [51, 55, 68, and references therein]. Human populations declined to a minimum during an extended hot and dry period between ca. 5000 to 4000 BP [68], followed by wetter and cooler conditions in the Middle Archaic period (ca. 4000 to 1000 BP) during which the hunting of larger game (bighorn sheep and deer) intensified [55, 62, 63, 71, 72]. Subsequently, droughts during the Medieval Climatic Anomaly (MCA, ca. 1200 to 650 BP) were a likely cause of declining populations during the last millennium BP, coincident with a shift to smaller prey [62, and references therein]. The atlatl was the dominant hunting weapon in the Great Basin region from about 8000 BP to about 1500 BP, and was subsequently replaced by the bow and arrow, which arrived in the region around 1600 BP and may have led to a depletion of bighorn populations [37, 53, 63, 73, 74].

Little is known about the ethnic and linguistic affiliation of the earliest pre-Numic residents of the region. The arrival of Northern Uto-Aztecan speakers from northern Mexico in the Great Basin region took place around 5000 BP, with Numic languages firmly established by 3000 BP [68]. Numic-speaking Shoshone and Paiute tribes occupied the area from around 600 BP through the Historic period [75]. The first contact with Euro-Americans in the Owens Valley is thought to have occurred in the 1830s, although indirect contact through trading had already begun in the 18[th] century [76]. The Owens Valley Natives were forcibly removed from their territory to Fort Tejon in 1863. A number of remaining Natives and returnees from Fort Tejon formed the basis of the present-day Native population (ca. 3000 persons) centered on the Lone Pine, Fort Independence, Big Pine, and Bishop reservations [77]. The archeological chronology of the Little Lake area has been summarized by Van Tilburg and Bretney [76] as follows: Lake Mojave (11,000–6000 BP), Little Lake (6000–3150 BP), Newberry (3150–1350

BP), Haiwee (1350–650 BP), Marana (650 BP– 1700 CE), and Historic periods (1700 CE– present).

## Geology

The study sites are on basaltic lava rocks, which overly a bedrock of Jurassic and Cretaceous granitic rocks that are about 165 and 100 million years old, respectively. The Big Pine volcanic field is of Quaternary age, with the youngest flows occurring near Aberdeen. Here, six distinct flow units have been dated to the late Pleistocene by [3]He and [36]Cl techniques [48, 50]. We measured the areal density of Mn and Fe on the flow surfaces and sampled varnish from four of these flow units at the same locations sampled by Vazquez and Woolford [50]: Units Qba (sample CLS-03, 40 ka), Qbbs (CLS-06, 27 ka), Qbac (CLS-04, 17 ka), and Qbtc (CLS-05, 17 ka). The precise locations of all measurement sites are given in Table 1. The complex surfaces of these lava flows provided the opportunity to make measurements on surfaces exposed to all cardinal directions and with inclinations from 0 to 90 degrees.

The measurements at Fossil Falls were made on Late Pleistocene vesicular basalts of the Red Hill flow of the Coso volcanic field [unit Qbr in reference 49]. This flow belongs to the most recent phases of activity of the Coso volcanic field and has been dated to about 60 ka [48]. The surface of this flow has been scoured by a late Pleistocene flood event of the Owens River, creating a fresh surface that has been dated to 16 ka by the [3]He technique [48]. This event represents the last time that Owens Lake discharged by way of the Owens River over Fossil Falls [78]. Here, pXRF measurements were made on un-scoured older surfaces, flood-scoured surfaces, and petroglyphs cut into both un-scoured and flood-scoured surfaces.

The Little Lake site (CA-INY-182) contains outcrops of both the Red Hill flow, which forms the ridge on the west side of Little Lake, and the Little Lake flow, which makes up the high and steep cliffs along the east side of the lake [78]. The latter flow originates from the Little Lake vent, located about 5 km east of the lake. It corresponds to unit Qbe in Duffield, Bacon [49] and has been K/Ar-dated to 140 ka BP. The pXRF measurements were made on intact varnish and petroglyphs of Red Hill basalt (Locus 4 and 8, see below) and Little Lake basalt (Atlatl Cliff and Locus 7). In addition, we made pXRF measurements at the site of the former Little Lake Hotel (LLH), 2 km S of the entrance to Little Lake Ranch, on outcrops of Red Hill basalt.

## Methods

**Portable X-Ray fluorescence spectrometry.** Our pXRF measurement technique has been described in detail in previous publications [17, 42] and will only be outlined briefly here.

**Table 1. Rock varnish sampling and measurement locations and substrate rock characteristics.**

| Locality | Sample/Site Code | Latitude [˚N] | Longitude [˚W] | Elevation [m asl] | Rock unit | Age [ka] |
|---|---|---|---|---|---|---|
| Big Pine Volcanic Field | CLS-03 | 36.978 | 118.272 | 1230 | Qba | 40 |
| Big Pine Volcanic Field | CLS-04 | 36.962 | 118.258 | 1187 | Qbac | 17 |
| Big Pine Volcanic Field | CLS-05 | 36.984 | 118.235 | 1180 | Qbtc | 17 |
| Big Pine Volcanic Field | CLS-06 | 36.945 | 118.241 | 1177 | Qbbs | 27 |
| Fossil Falls | FF | 35.970 | 117.906 | 1010 | Qbr | 60 |
| Fossil Falls, flood-scoured | FFS | 35.970 | 117.909 | 1010 | Qbr | 16 |
| Little Lake Hotel | LLH | 35.934 | 117.909 | 942 | Qbr | 60 |
| Little Lake, Atlatl Cliff | LLA | 35.958 | 117.904 | 973 | Qbe | 140 |
| Little Lake, Locus 4 | LL4 | 35.949 | 117.905 | 967 | Qbr | 60 |
| Little Lake, Locus 7 | LL7 | 35.942 | 117.905 | 964 | Qbe | 140 |
| Little Lake, Locus 8 | LL8 | 35.953 | 117.905 | 966 | Qbr | 60 |

Measurements were conducted using a Niton XL3 pXRF (Thermo Fisher Scientific) in the "mining" mode. The filter steps and integration periods were: "standard" 25 s, "low" 15 s, "high" 20 s, and "light" 25 s. The instrument is equipped with an X-ray source with an energy of 50 keV and a silver anode, and has a spot size of 8 mm in diameter. For quality control, the reference materials TILL-4 and FeMnOx-1 [GeoReM database, version 25; http://georem. mpch-mainz.gwdg.de; 79] were measured before and after each XRF measurement sequence. The measurement depth of the pXRF is dependent on the energy of the excitation and fluorescence photons, as well as on the composition (atomic number) of the analyte. The software in the instrument takes these factors into account when calculating the results. For the elements considered here, the measurement depth is of the order of a few tens to hundreds of microns. A total of 300 measurements were made, 158 on intact varnish surfaces, 120 on petroglyphs, and the rest for ancillary purposes, e.g., on freshly exposed bare basalt substrate. The measurements were typically made on several spots inside and adjacent to petroglyphs. For each spot, three to five replicate measurements were made by moving the pXRF a few mm or cm (depending on the size of the feature) within or near the petroglyph. Depending on the size and complexity of a given rock art element, one to five such spots were measured on each element. The locations of the spots are marked by arrows in S2 Fig. The measurement points on the adjacent intact varnish were chosen to be as close as possible to the petroglyph measurement spots, and to be as similar as possible in surface characteristics.

On the lava flows, close attention was paid to making measurements on original flow surfaces and avoiding surfaces formed by later fracturing and erosion. Six to 18 measurements were made on each lava flow by moving across the flow in more or less the same direction, and finding surfaces every few meters that were smooth enough to allow use of the pXRF, while also sampling a range of directions and inclinations. Since it is impossible to visually estimate varnish density on the black basalt, a selection bias is precluded.

While the measurement results from the bare basalt are valid as provided by the instrument in mass concentration units, the measurements on the rock varnishes were converted into areal density values, $D_{Mn}$, in units of µg cm$^{-2}$ using the calibration curve from Macholdt, Herrmann [80]. To correct for the underlying basalt element contribution, the Mn concentration of the unvarnished basalt was determined by conducting a measurement on a nearby freshly exposed rock surface and this value was subtracted from that measured on the varnished surface. The areal density of Fe ($D_{Fe}$) was calculated using the Mn calibration values and the Mn/Fe sensitivity ratio, and is thus subject to a greater uncertainty (estimated at about 20%). Since $D_{Mn}$ and $D_{Fe}$ vary substantially due to different growth and erosion conditions even within each rock art panel location, we also calculated the ratio of the measurements on the petroglyph surfaces to that on immediately adjacent intact varnish. This provides a normalized measure, called $N_{Mn}$ and $N_{Fe}$ (in %), which basically expresses the degree of re-varnishing on the petroglyph surface relative to the surrounding intact varnish. The measurement and data reduction techniques used were identical to those in Macholdt et al. [42] and are described in more detail there. Photographs of all petroglyph measurement locations are shown in the (S2 Fig).

Permission for the measurements at Little Lake was obtained from the owners (Little Lake Duck Club, Inc.). No permit was required for the collection of small rock samples and non-invasive measurements by pXRF on public lands. No samples of cultural heritage material were taken.

**Femtosecond LA-ICPMS.** The fs-LA-ICPMS measurements were carried out using a ThermoFisher Element 2 single-collector sector-field ICP-mass spectrometer combined with an ESI 200-nm femtosecond laser ablation system, NWRFemto. Laser ablation was conducted in a New Wave Large Format Cell using a He atmosphere. Subsequent to the ablation, the He

carrier gas was mixed with an Ar gas flow to transport the aerosols generated by ablation to the ICPMS. All measurements were conducted in medium mass resolution mode (2000) with flat-top peaks. The rock varnish measurements were executed, after pre-ablation with 80 μm s$^{-1}$ scan speed and a spot size of 65 μm, as in-situ line scans on the surfaces of unpolished slices cut perpendicular to the varnished rock surface. The operating parameters of the laser system during the measurements were: spot size: 40 μm, pulse repetition rate: 50 Hz, energy density: ca. 0.5 J cm$^{-2}$, scan speed: 1 μm s$^{-1}$, blank measurement 15 s, and washout time: 30 s. In addition to the line scan measurements, we also made spot measurements, where the laser was shot repeatedly onto the same spot on the varnish surface, thereby successively ablating deeper layers. Each shot (pulse repetition rate = 1 Hz) ablates ca. 50–100 nm, so that after 200–300 shots a depth of about 10–30 μm is reached. While this technique requires only minimal sample preparation (cutting a piece of the sample to fit into the ablation cell), it can be difficult to separate varnish and host rock on thinly and unevenly varnished surfaces. It also has lower sensitivity, so that only the main elements can be detected. The analytical error of the measurements is of the order of 2–6% for the elements measured.

Measurements with $MnO_2$ mass fractions of <2% were rejected as contamination from the underlying rock material. The reference glass GSE-1G (GeoReM database) was used for calibration. To normalize the data, the oxides of the major elements ($Na_2O$, MgO, $Al_2O_3$, $SiO_2$, $P_2O_5$, $K_2O$, CaO, $TiO_2$, $MnO_2$, and $Fe_2O_3$) were assumed to add up to 98 mass-%.

**Data analysis.**   Regression calculations were made using bivariate regression, which takes into account error in both the x and the y variables, using the Williamson-York Iterative bivariate fit algorithm [81].

## Results and discussion

### Rock varnish chemical composition

The chemical composition of the varnish was investigated in detail by fs-LA-ICPMS. The concentrations of major and trace elements in the CLS varnish samples are presented in Table 2. The dominant elements in all samples are Mn, Fe, Si, and Al, consistent with a composition dominated by Mn-Fe oxyhydroxides and clay minerals, as is typical of Type I rock varnishes [3, 7, 10]. The overall mean mass percentages of Mn and Fe in the varnishes are 4.9% and 12.9%, respectively, giving an average Mn/Fe mass ratio of 0.38. This ratio is considerably lower than the Mn/Fe ratios measured by pXRF (average 1.62) and may reflect the influence of signal from the basalt host rock, some of which may have been present in the relatively large laser spot (40 μm), and which has a much lower Mn/Fe ratio of 0.01 to 0.02. This is supported by the spot measurements, which showed much higher ratios (0.5 to 2.5) in the varnish layer (S3 Fig). Another factor is the great variability of the Mn oxyhydroxide concentration at the microscale, which makes the absolute concentrations measured by this technique subject to a considerable amount of random chance, depending on just where the line scan ends up on the varnish sample. The LA-ICPMS data are therefore more meaningfully interpreted in the form of the enrichment patterns discussed below.

The average Mn concentration of 4.9% corresponds to a MnO concentration of 6.6% in the varnish material. The highest observed Mn concentration, in CLS05, was 11.3%, corresponding to 15.2% MnO. These values fall at the low end of the range reported by Broecker and Liu [82] and are typical of varnish formed under arid conditions at annual precipitation rates around 50–150 mm, which suggests that varnish formation in the study region took place predominantly under arid conditions. Further discussion of the Mn/Fe ratios, including the pXRF results from the other sites in this study, is presented in the section on the areal densities of Mn and Fe below.

**Table 2. Elemental composition of the rock varnish on the Big Pine volcanic field basalts at Aberdeen, CA, as measured by fs-LA-ICPMS (in µg g$^{-1}$).** The standard deviations reflect mostly the variability of trace elements in the varnish and not the analytical error, which is of the order of 2–6% for the elements measured.

| Element | CLS-04 (n = 35) | | CLS-05 (n = 30) | | CLS-06 (n = 112) | |
|---|---|---|---|---|---|---|
| | Average | Std. Dev. | Average | Std. Dev. | Average | Std. Dev. |
| B | 52 | 84 | 78 | 38 | 87 | 130 |
| Na | 17900 | 5800 | 19100 | 11100 | 10000 | 3400 |
| Mg | 30100 | 14100 | 20800 | 10900 | 24600 | 9600 |
| Al | 88300 | 19900 | 72900 | 11100 | 92500 | 11800 |
| Si | 182000 | 22000 | 163000 | 30000 | 171000 | 18000 |
| P | 3600 | 1500 | 2900 | 800 | 4600 | 2000 |
| K | 7300 | 4900 | 21700 | 9700 | 8900 | 8500 |
| Ca | 83400 | 20400 | 43400 | 19900 | 52900 | 19400 |
| Ti | 11000 | 2300 | 14100 | 5700 | 13300 | 2700 |
| Mn | 31100 | 7900 | 79300 | 51200 | 47200 | 17700 |
| Fe | 104000 | 32000 | 133000 | 33000 | 135000 | 31000 |
| Co | 271 | 103 | 506 | 311 | 305 | 125 |
| Ni | 40 | 60 | 119 | 55 | 71 | 73 |
| Rb | 18.8 | 15.2 | 88 | 37 | 44 | 26 |
| Sr | 1300 | 500 | 1100 | 400 | 800 | 300 |
| Y | 47 | 20 | 97 | 42 | 71 | 18 |
| Cs | 2.2 | 2.3 | 8.1 | 5.3 | 6.0 | 4.9 |
| Ba | 3100 | 800 | 10200 | 6800 | 4200 | 2300 |
| La | 100 | 34 | 250 | 158 | 146 | 48 |
| Ce | 570 | 156 | 2593 | 2112 | 1251 | 485 |
| Pr | 25 | 8 | 56 | 30 | 35 | 11 |
| Nd | 93 | 40 | 190 | 98 | 129 | 45 |
| Sm | 20 | 13 | 33 | 19 | 26 | 10 |
| Eu | 5.6 | 6.7 | 7.1 | 2.8 | 5.1 | 2.3 |
| Gd | 14.4 | 13.6 | 24.2 | 11.7 | 20.3 | 9.5 |
| Tb | 2.3 | 2.2 | 3.6 | 1.7 | 3.0 | 1.4 |
| Dy | 12.5 | 9.8 | 18.8 | 10.7 | 17.2 | 7.3 |
| Ho | 2.1 | 1.5 | 3.6 | 1.6 | 3.0 | 1.3 |
| Er | 6.6 | 6.4 | 11.1 | 5.9 | 11.4 | 6.0 |
| Tm | 1.2 | 1.3 | 1.2 | 0.9 | 1.3 | 0.8 |
| Yb | 6.1 | 5.4 | 11.8 | 6.8 | 9.3 | 5.6 |
| Lu | 1.4 | 2.6 | 1.7 | 0.8 | 1.3 | 0.8 |
| Pb | 1500 | 1200 | 1100 | 1000 | 3300 | 2600 |
| Th | 78 | 62 | 148 | 76 | 154 | 73 |
| U | 38 | 17 | 66 | 39 | 92 | 34 |
| Zr | 362 | 135 | 501 | 185 | 407 | 113 |
| Cu | 230 | 139 | 249 | 122 | 1096 | 2711 |
| Cr | 72 | 72 | 56 | 59 | 164 | 237 |
| Zn | 138 | 215 | 391 | 253 | 455 | 388 |
| V | 214 | 72 | 470 | 108 | 334 | 90 |

For further discussion, the elemental composition data are presented in Fig 1A in the form of enrichment factors against the average upper continental crust (UCC) composition [83]. In this figure, the results of our previous measurements on Type I varnish are shown for comparison. Generally, the enrichment factors against UCC composition are in good agreement with

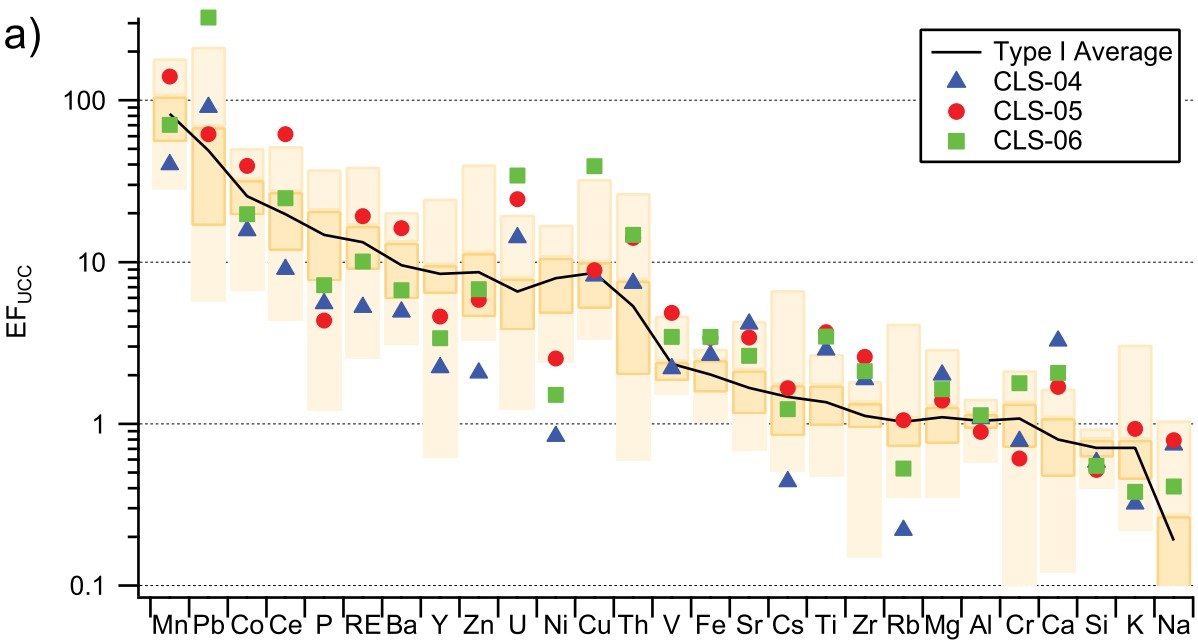

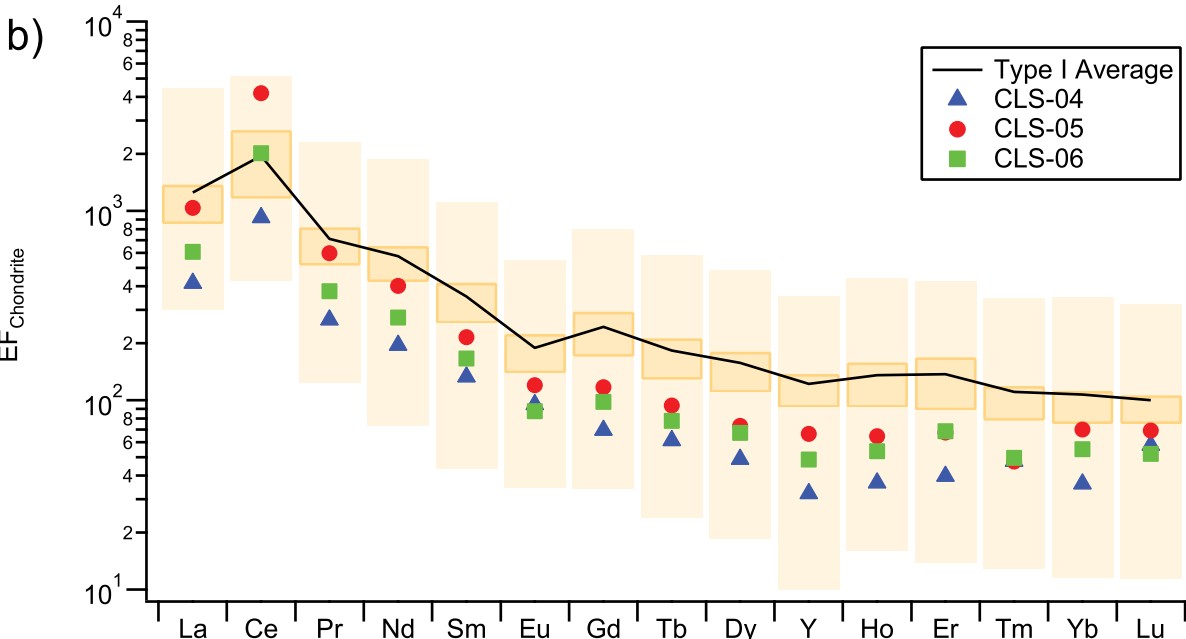

**Fig 1. Chemical composition of the rock varnish on the lava flows at Aberdeen, CA.** (a) Elemental composition expressed as enrichment factors vs. average upper continental crust. (b) Rare earth element and yttrium enrichment factors vs. carbonaceous chondrite C1 composition. The black line represents the average of our previous data from Type I varnish; the color bars show the range of our previous measurements, with the darker color indicating the interquartile range. (RE stands for the sum of rare earth elements and yttrium).

published values for the elements typically enriched in varnish, e.g., Mn, Pb, Ba, Ce, and Co [3, 10, 12, 16, 18, 84, 85].

The main varnish elements, Mn and Fe, are both strongly enriched in the CLS varnish samples. The Fe enrichment factors are slightly above average, and consequently the Mn/Fe ratios are relatively low, in the range typical for varnish formed under arid conditions. Silica is more depleted than the Type 1 average, possibly because detrital quartz grains are less abundant in this area, which is dominated by basaltic host rocks, than in most of our other locations, which had sandstone and gneiss as host rocks. The above-average Ca content may be due to the presence of some Ca carbonate or Ca-containing clay minerals of the smectite group, which are common basalt weathering products [86–88].

Some unusually high enrichments stand out in this plot, particularly for Pb, Ce, Cu, U, and Th. In contrast, the P and Ni enrichments are unusually low. Cobalt is highly enriched, but still within the typical range for Type I. We have previously observed a similar enrichment pattern, characterized by very high Pb, Ce, Th and U enrichments, but low Ni and Co enrichments relative to the Type I average, in samples from other locations in the Mojave Desert [7, 18]. This consistency among sites in the same region is likely related to a regional similarity of dust composition and enrichment processes. In particular, the high Th and U enrichments may be related to the presence in the region of a high proportion of felsic intrusive and extrusive igneous rocks (e.g., the Long Valley rhyolites, and the Southern California and Sierra Nevada batholiths) with strongly elevated U and Th concentrations [89–91].

The most extreme Pb enrichment is observed in CLS-06. This may be related to the fact that this sample was collected very close (ca. 100 m) to a highway with a high traffic density, and may have accumulated automotive Pb. The lead enrichment is highest at the surface, but reaches well into the varnish, indicating the lead absorption to the varnish occurs not only at the immediate surface but also throughout a layer of significant thickness. Lead may also have been mobilized and redistributed during diagenetic processes within the varnish [5, 13, 92].

The rare earth element enrichment pattern is shown in Fig 1B in the form of enrichment ratios vs. the composition of carbonaceous chondrites [93] (for this discussion, we include yttrium [Y] with the REE, using the abbreviation REY). We find the light REE (LREE) up to one order of magnitude more strongly enriched than the heavy REY (HREY), with a gradual decrease of the EFs from La to Lu. Three anomalies are present in this series: A strong positive Ce anomaly, a distinct negative Eu anomaly, and a slight negative Y anomaly. These trends and anomalies are in overall agreement with previous studies [7, 12, 14, 16, 18, 94–96]. The REY enrichments in varnish have been attributed to preferential absorption of the REY to Mn oxyhydroxides in the course of the leaching and absorption processes that lead to varnish formation [16–18].

The Ce anomaly results from the fact that in the oxidizing environment, which exists on the surface of the Mn oxyhydroxides, Ce is in the highly insoluble and sorption-prone $Ce^{4+}$ oxidation state, whereas the other REE are in the more soluble 3+ oxidation state (Eu and Y may be partially also in the 2+ oxidation state). Cerium thus accumulates irreversibly over time and a strong Ce anomaly suggests slow growth [7, 96]. Fig 1B shows that, whereas all other EFs in our samples are below average, the Ce EFs are clustered around the Type I varnish average. The Ce anomaly (defined as the ratio between the enrichment factor of Ce and the average of the enrichment factors of La and Pr) in the Aberdeen lava flows varnishes ranges between 2.7 and 5.1, with the latter being the highest value we have found anywhere in rock varnishes. Similar high Ce enrichments (3–4) have been found in the Mojave Desert and Death Valley by Thiagarajan and Lee [16].

In contrast to Ce, Eu typically shows a negative anomaly in rock varnish (Fig 1B). This is likely related to the fact that Eu can also exist in the $Eu^{2+}$ oxidation state, which is soluble and less prone to adsorption. Any $Eu^{2+}$ released during the leaching stage would thus have to be first oxidized to $Eu^{3+}$ before it can be absorbed to the oxyhydroxides. A similar explanation

might apply to the slight negative Y anomaly, as this element can also exist in the $Y^{3+}$ and $Y^{2+}$ oxidation states [18]. Bau, Schmidt [96] proposed a classification of marine Fe-Mn oxyhydroxide deposits based on plots of the Ce anomaly vs. Nd concentration and Ce anomaly vs. Y anomaly, and related the resulting classification to the proposed REY scavenging mechanism. The results from our measurements plot in the field of hydrogenetic Fe-Mn crusts, suggesting that the same scavenging mechanism, i.e., scavenging from aqueous solution onto Mn/Fe oxyhydroxides also applies to our rock varnish samples.

The general decrease of enrichment from the LREE to the HREY has been suggested to result from the differential behavior of the distribution coefficients of these elements between oxyhydroxides and clay minerals [18]. While the distribution coefficients between solution and solid phase are similar between heavy and light REY for the oxyhydroxides, they increase by about one order of magnitude from La to Lu for the clay minerals, so that the heavy REY are less prone to leaching from the clay minerals and reprecipitation in the oxyhydroxides than the LEE. In addition, the HREY form stronger complexes with carbonate and organic ligands and are thus more likely to remain in solution.

In conclusion, the enrichment pattern of trace elements and REY in the varnish is consistent with a formation mechanism, in which they are leached under initially slightly acidic conditions typical of hydrometeors (rain or dew), followed by precipitation of Mn-oxyhydroxides and scavenging of trace elements by these Mn-oxyhydroxides when the pH increases as the moisture on rock surfaces reacts with carbonate and silicate minerals, consuming $H^+$ ions and releasing mineral cations [14, 16, 18, 97].

## Areal density of manganese and iron in the rock varnish

The results of our pXRF measurements on rock varnish are shown as a scatter plot of surface densities, $D_{Mn}$ vs. $D_{Fe}$, in Fig 2A, and the normalized surface densities, $N_{Mn}$ and $N_{Fe}$, are presented in Fig 2B. The corresponding summary statistics are given in Table 3, and the more detailed statistics for the Aberdeen lava flows, the intact varnish surrounding the petroglyphs, and the petroglyphs are presented in Tables 4–6. The average $D_{Mn}$ from all measurements is $350\pm310$ µg cm$^{-2}$, the large standard deviation reflecting the wide range of varnish coatings, from some very recent, light petroglyph surfaces to visually very dark surfaces. The average $D_{Mn}$ value on the petroglyphs ($160\pm170$ µg cm$^{-2}$) is only about 30% of that on the adjacent intact varnish ($550\pm290$ µg cm$^{-2}$) and about one-half of that on the Late Quaternary Aberdeen basalts ($300\pm150$ µg cm$^{-2}$). In spite of the large variability of the Mn densities, the difference between the Aberdeen values and those of intact varnish from LL and FF is statistically significant based on a t-test (p<0.0001). A more detailed analysis of the intact varnish data suggests this difference may be related to the age of the surfaces, as the mean $D_{Mn}$ increases from 340 $\pm190$ µg cm$^{-2}$ on the Red Hill flow surface that had been scoured by flooding until 16 ka BP, through $560\pm310$ µg cm$^{-2}$ on the un-scoured 60 ka Red Hill basalt, to $750\pm280$ on the 140 ka Little Lake flow (Table 3).

The intact varnish densities from all our sites are much larger than the values we had measured previously in Saudi Arabia, with $105\pm55$ µg cm$^{-2}$ in the Hima region [42] and $156\pm94$ µg cm$^{-2}$ in the Ha'il region [17]. There are many potential reasons for this difference, including differences in age, rock substrate, climate, dust availability, rainfall, and others. Present-day rainfall is similar in the Owens/Rose Valley study area and our Arabian sites (125–175 mm vs. ca. 130 mm annually, respectively), and both regions have experienced lengthy periods of higher rainfall during the Holocene. This makes differences in rainfall unlikely as an explanation for the observed differences in $D_{Mn}$, especially in view of the weak relationship between rainfall and Mn content shown by Broecker and Liu [82]. Dust fluxes in the Owens/Rose

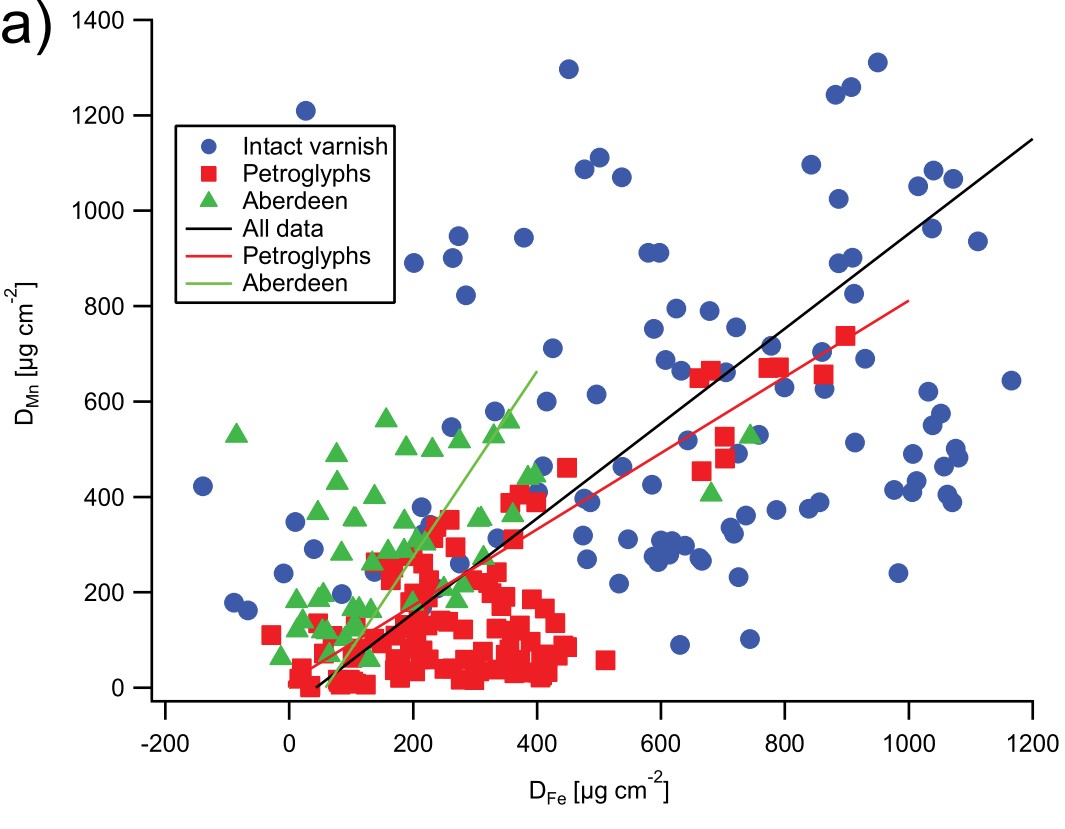

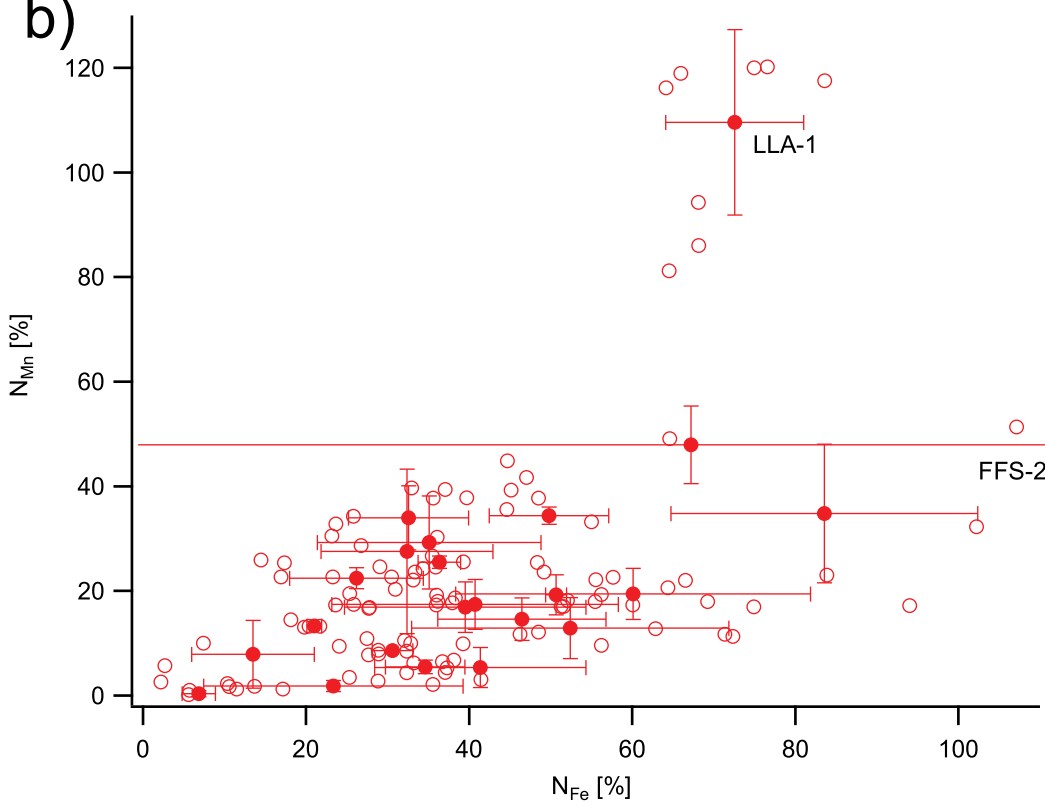

**Fig 2. Areal density of Mn vs. areal density of Fe on the rock varnishes in the Owens/Rose Valley study area.** (a) Areal density of Mn ($D_{Mn}$) vs. areal density of Fe ($D_{Fe}$), (b) normalized areal densities, $N_{Mn}$ vs. $N_{Fe}$, on the petroglyphs. The lines represent linear regressions; the regression parameters can be found in Table 3. Negative $D_{Fe}$ values are a consequence of the relatively large uncertainty resulting from the subtraction of the host rock Fe contribution from the Fe signal.

Valley are lower than those in Arabia both in the Pleistocene and the Holocene [98–100], ruling this variable out as a possible explanation. We speculate that the most likely reason for the greater $D_{Mn}$ in the present study area is the fact that the rock substrate, basalt, is more resistant to weathering than the sandstone substrate at our petroglyph sites in Arabia and thus enables a longer period of varnish accumulation. Consistent with this hypothesis is the observation that high $D_{Mn}$ values were also found on very resistant silica- and hematite-cemented sandstones that formed desert pavements in Saudi Arabia (410–670 µg cm$^{-2}$; M. O. Andreae, unpublished data, 2015).

There are, unfortunately, only very few published values of $D_{Mn}$ from North America to which we can compare our results. Measurements of intact varnish on sandstones in Utah by a technique similar to ours yielded much lower $D_{Mn}$ values: 24–87 µg cm$^{-2}$ [39]. In a study of rock varnish on small clasts from piedmont slopes in the Mojave Desert, Reneau [34] determined median values of 104 µg cm$^{-2}$ on mid- to late-Holocene surfaces, 130 to 220 µg cm$^{-2}$ on early- to mid-Holocene surfaces, and 90 to 220 µg cm$^{-2}$ on Pleistocene surfaces. Interestingly, these values agree reasonably well with our results, especially with those from the petroglyphs, which are also presumed to be of Holocene age [36, 51].

The mean Fe areal densities show a similar trend, ranging from 170±200 µg cm$^{-2}$ on the flood-scoured Red Hill basalt to 890±240 µg cm$^{-2}$ on the Little Lake basalt (Table 3). The petroglyph surfaces have a mean $D_{Fe}$ of 300±180 µg cm$^{-2}$. The overall average $D_{Fe}$ of the intact varnishes is 630±320 µg cm$^{-2}$, about 2–3 times higher than our results from Saudi Arabia (330±80 µg cm$^{-2}$ at Hima and 185±121 µg cm$^{-2}$ at Ha'il). The overall mean Mn/Fe mass ratio is 1.01±1.00, with the individual surface types ranging from 0.57±0.51 on the petroglyph surfaces

**Table 3. Mn and Fe areal density, normalized areal density, Fe vs. Mn correlation, and Mn/Fe ratios from pXRF measurements on rock varnish at Aberdeen volcanic area, Fossil Falls (FF), Little Lake (LL), and Little Lake Hotel (LLH).** [Avg.: arithmetic average, S.D.: standard deviation, S.E.: standard error].

| Site | N | $D_{Mn}$ | | $D_{Fe}$ | | Slope Mn/Fe | | Intercept | | $R^2$ | Mn/Fe | |
|---|---|---|---|---|---|---|---|---|---|---|---|---|
| | | Avg. | S.D. | Avg. | S.D. | Avg. | S.E. | Avg. | S.E. | | Avg. | S.D. |
| Aberdeen lava flows | 45 | 300 | 150 | 180 | 160 | 1.64 | 0.28 | 12.3 | 52.3 | 0.31 | 1.62 | 0.24 |
| Flow surfaces at Rose Valley sites FF, LL, and LLH | | | | | | | | | | | | |
| Red Hill | 58 | 560 | 310 | 620 | 230 | - - - | - - - | - - - | - - - | - - - | 0.89 | 0.60 |
| Red Hill scoured | 16 | 340 | 200 | 150 | 200 | 1.81 | 0.62 | -138 | 190 | 0.39 | 1.19 | 0.91 |
| Little Lake basalt | 26 | 750 | 280 | 890 | 240 | 1.90 | 0.72 | -939 | 646 | 0.05 | 0.84 | 0.39 |
| All intact varnish | 103 | 550 | 290 | 630 | 320 | 1.01 | 0.16 | -47 | 109 | 0.04 | 0.86 | 0.63 |
| Petroglyphs | 107 | 160 | 170 | 300 | 180 | 0.93 | 0.08 | -114 | 26 | 0.43 | 0.57 | 0.51 |
| All data | 262 | 350 | 310 | 400 | 310 | 1.00 | 0.07 | -43 | 32 | 0.28 | 1.01 | 1.00 |
| Normalized densities on petroglyphs | | | | | | | | | | | | |

| | N | $N_{Mn}$ | | $N_{Fe}$ | | Slope Mn/Fe | | Intercept | | $R^2$ | | |
|---|---|---|---|---|---|---|---|---|---|---|---|---|
| | | Avg. | S.D. | Avg. | S.D. | Avg. | S.E. | Avg. | S.E. | | | |
| Individual measurements | 106 | 26.6 | 28.4 | 41.6 | 26.6 | 1.16 | 0.18 | -22 | 6.6 | 0.22 | | |
| Same, w/o outliers[a] | 94 | 18.1 | 11.6 | 37.8 | 19.0 | 0.32 | 0.06 | 5.9 | 2.5 | 0.11 | | |
| Element averages | 22 | 23.3 | 22.8 | 41.6 | 18.5 | 1.40 | 0.32 | -35 | 14 | 0.34 | | |
| Same, w/o outliers[a] | 20 | 17.7 | 10.8 | 38.8 | 16.9 | 0.55 | 0.14 | -3.2 | 6.1 | 0.24 | | |

[a]) Data from FFS-2 and LLA-1 removed as outliers

**Table 4. Mn and Fe areal density, radiometric age, and apparent accumulation rates.** Mn and Fe areal density, radiometric age, and apparent Mn and Fe accumulation rates on the different flow units of the Aberdeen lava flows of the Big Pine volcanic field, the Red Hill and Little Lake lava flows, and the 16-ka flood-scoured surface of the Red Hill flow at Fossil Falls.

| Flow unit | Age | N | $D_{Mn}$ | | | $D_{Fe}$ | | | Mn accumulation rate | | Fe accumulation rate | |
|---|---|---|---|---|---|---|---|---|---|---|---|---|
| | | | Avg. | S.D. | CV | Avg. | S.D. | CV | Avg. | S.E. | Avg. | S.E. |
| | [ka] | | [µg cm$^{-2}$] | | | [µg cm$^{-2}$] | | | [µg cm$^{-2}$ ka$^{-1}$] | | [µg cm$^{-2}$ ka$^{-1}$] | |
| CLS-03 | 40 | 15 | 310 | 130 | 42% | 260 | 170 | 66% | 7.7 | 0.9 | 6.6 | 1.2 |
| CLS-04 | 17 | 11 | 350 | 170 | 48% | 190 | 160 | 87% | 20.6 | 2.7 | 11.0 | 2.6 |
| CLS-05 | 17 | 18 | 250 | 150 | 62% | 93 | 59 | 63% | 14.6 | 2.4 | 5.5 | 0.9 |
| CLS-06 | 27 | 6 | 350 | 130 | 37% | 190 | 103 | 55% | 13.1 | 1.3 | 7.0 | 1.0 |
| Average | | | | | 47% | | | 68% | | | | |
| Red Hill, scored | 16 | 15 | 340 | 200 | 60% | 150 | 200 | 136% | 20.9 | 3.4 | 9.1 | 3.3 |
| Red Hill flow | 60 | 58 | 560 | 310 | 56% | 620 | 230 | 37% | 9.3 | 1.4 | 10.4 | 1.0 |
| Little Lake flow | 140 | 26 | 750 | 180 | 24% | 890 | 240 | 27% | 5.4 | 0.3 | 6.4 | 0.5 |

to 1.62±0.24 on the Aberdeen lava flows (Table 3). The observed Mn/Fe ratios of <1 in the varnish on the Holocene petroglyphs and higher ratios on the late Pleistocene lava flows are consistent with the microstratigraphic analyses of Liu and coworkers [22, 101, 102], who found

**Table 5. Mn and Fe areal densities ($D_{Mn}$ and $D_{Fe}$) of the intact rock varnish surrounding the petroglyphs.** The labels in the Element column correspond to the rock art elements in Table 6.

| Element | N | $D_{Mn}$ | | | $D_{Fe}$ | | |
|---|---|---|---|---|---|---|---|
| | | Avg. | S.D. | CV | Avg. | S.D. | CV |
| | | [µg cm$^{-2}$] | | | [µg cm$^{-2}$] | | |
| LLH-1i | 4 | 330 | 71 | 21% | 540 | 97 | 18% |
| LLH-2i | 4 | 530 | 59 | 11% | 300 | 90 | 30% |
| LLH-3i | 4 | 520 | 200 | 39% | 720 | 210 | 29% |
| FFS-1i | 3 | 660 | 220 | 33% | 420 | 150 | 36% |
| FFS-2i | 7 | 260 | 96 | 37% | --- | --- | --- |
| FFS-F | 5 | 240 | 57 | 24% | 220 | 50 | 23% |
| FF-1&2i | 7 | 320 | 73 | 23% | 750 | 130 | 18% |
| FF-3i | 5 | 370 | 140 | 38% | 780 | 190 | 24% |
| FFS-L | 7 | 910 | 300 | 33% | 300 | 160 | 53% |
| LLA-1i | 4 | 560 | 54 | 10% | 1030 | 19 | 2% |
| LLA-2i | 4 | 1000 | 79 | 8% | 970 | 82 | 8% |
| LLA-3i | 5 | 1030 | 160 | 16% | 1000 | 92 | 9% |
| LLA-4i | 4 | 480 | 120 | 24% | 1100 | 48 | 4% |
| LL8-1i | 3 | 380 | 95 | 25% | 300 | 110 | 35% |
| LL8-2i | 2 | 400 | 120 | 30% | 600 | 180 | 30% |
| LL8-3i | 4 | 710 | 130 | 18% | 760 | 180 | 24% |
| LL7-1i | 4 | 450 | 32 | 7% | 1040 | 36 | 3% |
| LL7-2i | 5 | 750 | 210 | 28% | 620 | 100 | 16% |
| LL9-3i | 3 | 340 | 19 | 6% | 720 | 13 | 2% |
| LL7-4i | 2 | 96 | 9 | 9% | 690 | 80 | 12% |
| LL4-1i | 7 | 910 | 210 | 23% | 610 | 55 | 9% |
| LL4-2i | 3 | 740 | 63 | 9% | 570 | 72 | 13% |
| Average | | | | 21% | | | 19% |

**Table 6. Mn and Fe areal densities ($D_{Mn}$ and $D_{Fe}$) and normalized densities ($N_{Mn}$ and $N_{Fe}$) on the petroglyphs (rock art elements).**

| Element | Motif | N | $D_{Mn}$ | | | $D_{Fe}$ | | | $N_{Mn}$ | | | $N_{Fe}$ | | |
|---|---|---|---|---|---|---|---|---|---|---|---|---|---|---|
| | | | Avg. | S.D. | CV | Avg. | S.D. | CV | Avg. | S.D. | S.E. | Avg. | S.D. | S.E. |
| | | | [µg cm$^{-2}$] | | | [µg cm$^{-2}$] | | | [%] | | | [%] | | |
| LLH-1 | Anthropomorph | 5 | 65 | 16 | 25% | 330 | 120 | 36% | 19 | 4.9 | 16% | 60 | 22 | 20% |
| LLH-2 | Coso bighorn (III) | 3 | 180 | 70 | 38% | 280 | 63 | 23% | 35 | 13 | 28% | 84 | 19 | 26% |
| LLH-3 | Metate | 3 | 10 | 6 | 58% | 170 | 110 | 68% | 1.8 | 1.1 | 49% | 23 | 16 | 52% |
| FFS-1 | Bighorn sheep (I) | 4 | 180 | 104 | 57% | 140 | 45 | 33% | 28 | 16 | 38% | 32 | 10 | 28% |
| FFS-2 | Atlatl | 4 | 130 | 19 | 15% | 29 | 95 | 325% | 48 | 7.4 | 23% | 67 | 98 | - - - |
| FF-1 | Coso bighorn (III) | 5 | 92 | 28 | 30% | 260 | 100 | 39% | 29 | 8.9 | 19% | 35 | 14 | 21% |
| FF-2 | Coso bighorn (III) | 4 | 55 | 15 | 27% | 300 | 130 | 43% | 17 | 4.8 | 21% | 41 | 18 | 27% |
| FF-3 | "Medicine bag" | 4 | 54 | 15 | 28% | 360 | 81 | 22% | 15 | 4.0 | 27% | 46 | 10 | 19% |
| LLA-1 | Curvilin. abstract | 9 | 610 | 99 | 16% | 750 | 87 | 12% | 110 | 18 | 7% | 73 | 8.4 | 4% |
| LLA-2 | Atlatl | 6 | 220 | 20 | 9% | 250 | 79 | 31% | 22 | 2.0 | 5% | 26 | 8.2 | 14% |
| LLA-3 | Atlatl | 10 | 350 | 63 | 18% | 330 | 74 | 23% | 34 | 6.1 | 8% | 33 | 7.4 | 8% |
| LLA-4 | Atlatl (repecked) | 6 | 26 | 6 | 24% | 380 | 53 | 14% | 5.5 | 1.3 | 15% | 35 | 4.9 | 7% |
| LL8-1 | Metate | 3 | 21 | 15 | 72% | 120 | 39 | 31% | 5.4 | 3.8 | 54% | 41 | 13 | 33% |
| LL8-2 | Metate | 3 | 2 | 2 | - - - | 41 | 12 | 30% | 0.4 | 0.6 | - - - | 6.8 | 2.0 | 30% |
| LL8-3 | Coso bighorn (III) | 9 | 56 | 47 | 83% | 93 | 60 | 64% | 7.9 | 6.5 | 30% | 14 | 7.5 | 24% |
| LL7-1A | Bighorn sheep (I) | 2 | 59 | 0 | 0% | 220 | 10 | 4% | 13 | 0.0 | 7% | 21 | 0.9 | 6% |
| LL7-1B | Bighorn sheep (I) | 2 | 38 | 0 | 1% | 320 | 26 | 8% | 8.6 | 0.1 | 7% | 31 | 2.5 | 9% |
| LL7-2 | PB Anthropom. | 10 | 97 | 44 | 45% | 320 | 120 | 37% | 13 | 5.8 | 18% | 52 | 20 | 13% |
| LL7-3 | "Bear paw" | 3 | 65 | 13 | 20% | 370 | 9 | 3% | 19 | 3.8 | 15% | 51 | 1.3 | 2% |
| LL7-4 | Anthropomorph | 2 | 33 | 2 | 5% | 340 | 50 | 15% | 34 | 1.7 | 10% | 50 | 7.3 | 19% |
| LL4-1 | Bighorn sheep (II) | 9 | 140 | 31 | 22% | 240 | 73 | 30% | 17 | 4.8 | 11% | 40 | 15 | 11% |
| LL4-2 | Rectilinear abstract | 3 | 190 | 9 | 5% | 210 | 15 | 7% | 26 | 1.2 | 7% | 36 | 2.6 | 10% |
| Average | | | | | 28% | | | 27%[a] | | | 20% | | | 18%[a] |

[a]) without FFS-2

that the Holocene was characterized by a yellow layer (in microscope slides) with Mn/Fe <1 formed under arid conditions, while varnish from the last glacial period contained black layers with Mn/Fe up to ~4 reflecting wetter periods. On considerably older lava flow surfaces of the Cima volcanic field (15 to 460 ka), Reneau, Raymond [103] also found high Mn/Fe ratios (averaging around 2).

The Mn/Fe ratios in the Owens/Rose Valley varnish are similar to our intact varnish Mn/Fe ratios of 0.91±0.64 at Ha'il (Saudi Arabia), and to measurements in Nevada and Utah (1.00 ±0.38) [95] and the Negev Desert, Israel (1.31±0.23) [94]. In contrast, they are much higher than our values from Hima (0.32±0.16) in Saudi Arabia and previous published Mn/Fe ratios from California: 0.22±0.08 in Death Valley and the Mojave Desert [16], 0.09 to 0.24 at another Mojave Desert location [104], and 0.09 to 0.71 at two other sites in the Mojave [105]. The reasons for these differences are not clear; they may be related to differences in climatic wetness during varnish formation [82, 101, 102, 106].

In our studies on Arabian varnishes, we had observed a positive iron intercept in regression analyses of $D_{Fe}$ vs. $D_{Mn}$, which we suggested to be due to the presence of an Fe oxyhydroxide layer either at the base or the top of the varnish. Such an Fe oxyhydroxide layer has been suggested to act as a catalyst for the formation of the Mn oxyhydroxides [17, 20]. In contrast, regression analyses on the data in this study showed no consistent pattern (Table 3). This difference is likely due to the different characteristics of the rock substrate. In the more porous

weathering rind of the sandstones studied in Arabia, a brown Fe oxyhydroxide layer is often clearly visible penetrating as much as several mm into the substrate [for images see 42]. Such a layer was not seen in the more resistant basalt substrate in this study.

In order to test whether the varnish deposition depends on the cardinal orientation or slope of the rock surface, we made measurements on the Aberdeen lava flows on surfaces facing all four cardinal directions and with slopes with inclinations from 0˚ to 80˚. Since there were no significant differences in the $D_{Mn}$ values between the four different flow units, we grouped all data into four cardinal orientations, which gave the following means and standard deviations (all in $\mu g\ cm^{-2}$): North 2.18±0.60 (n = 14), East 2.50±0.54 (n = 14), South 1.89±0.61 (n = 9), and West 1.78±0.58 (n = 18). T-tests indicate that these differences are not statistically significant, in agreement with Lytle, Lytle [38], who had also found no difference between rock faces with different orientation on varnished boulders from Idaho. It is, however, possible that in more mesic settings, especially with rain or dew coming from a preferred direction, cardinal orientation may play a more significant role.

A plot of the $D_{Mn}$ values against the inclination of the rock surface suggested a significant negative correlation (Panel (a) in S4 Fig), as could be expected given the generally accepted idea that the varnish is derived from the processing of deposited aeolian dust. Lytle, Lytle [38] had also observed such a relationship and proposed a correction by dividing the observed Mn areal density by the cosine of the slope inclination to obtain a normalized density. Since this implies the counterfactual result that there should be no varnish on vertical surfaces, this correction is obviously too strong. We tested several possibilities, including a linear correction, and found that the best correction could be achieved with an "attenuated" cosine correction:

$$D^0_{Mn} = D_{Mn} * 1/\cos(a * I)$$

where $D_{Mn}$ is the measured Mn areal density, $D^0_{Mn}$ the value normalized to an inclination, *I*, of zero (horizontal), and *a* is an attenuation factor (<1) that prevents the correction from reaching excessive values when *I* approaches 90˚. We found that with our data set, a value of *a* = 0.853 reduced the regression slope to zero and provided an adequate, although not perfect, correction (Panel (b) in S4 Fig).

## Normalized areal density of manganese and iron on the petroglyph surfaces

As a metric for the degree to which varnish has regrown on a petroglyph surface after removal of the original varnish by pecking or abrasion to create the petroglyph, we developed in our previous work the concept of the normalized Mn areal density, $N_{Mn}$, defined as the areal density of Mn on a petroglyph surface divided by that on an adjacent intact rock varnish surface, expressed in percent [17, 42, 80]. This value can be considered as the regrowth percentage of the varnish following its creation by removal of the preexisting varnish to create the rock art. (We avoid the use of the terms "patination" or "repatination", frequently found in the literature, as rock varnish is strictly speaking not a patina, i.e., an oxidation or weathering product of the substrate, but rather a coating derived from an external source.) This normalization adjusts for the considerable variability of varnish thickness and growth on scales comparable to the distance between the measurement points on the petroglyph and the points on the adjacent intact varnish, usually a few cm or tens of cm, depending on the size of the petroglyph. It has the advantage of eliminating the effect of the inclination of the rock surface, since the inclination of the petroglyph and that of the adjacent intact surface is essentially the same. It also ensures that the petroglyph surface has the same microclimate, substrate characteristics, etc. as the reference intact varnish. Variability on the size scale of the petroglyph itself is taken into

account by making multiple measurements within and adjacent to a given petroglyph, and variability on the microscale is averaged over by the spot size (8 mm) of the pXRF measurement.

The scatter plot of $N_{Mn}$ vs. $N_{Fe}$ (Fig 2B) shows that, with the exception of the measurements of one petroglyph (LLA-1), the $N_{Mn}$ values fall between 0 and 57%, while the $N_{Fe}$ values reach from near 0 to 107% (with the exception of two very erratic measurements from Fossil Falls on the atlatl element FFS-2). The very high $N_{Mn}$ from image LLA-1 at Atlatl Cliff and the very noisy $N_{Fe}$ measurements on FFS-2 from Fossil Falls will be discussed in detail below. The correlations between $N_{Mn}$ and $N_{Fe}$ are statistically significant, but represent only a minor fraction of the variance, as indicated by their low $r^2$ values, both when individual measurements are regressed ($r^2 = 0.22$) and when the averages from the measurements on the same petroglyph are used ($r^2 = 0.34$) (Table 3). When the above-mentioned outlier values are removed, the $r^2$ values decrease further, indicating that there is no meaningful relationship between $N_{Fe}$ and $N_{Mn}$, similar to what we had previously observed in our Arabian studies. This is not altogether unexpected, since Fe and Mn tend to be enriched in separate layers in the rock varnishes, which have been interpreted as representing contrasting depositional environments [4, 7, 102, 107]. Like in our previous work, we will focus our subsequent discussion of varnish growth in the petroglyphs on $N_{Mn}$.

## Variability of the Mn and Fe areal densities

The variability of varnish deposits on a given rock surface or within a petroglyph element can be considerable, depending on many factors, such as substrate resistance to weathering, exposure, climate, etc. For a summary of these factors, see the table in the (S1 Table). This variability is often emphasized in the literature, but rarely expressed in quantitative terms. Here, to quantify this variability, we calculated the coefficients of variation (CV), i.e., the ratio of the standard deviation over the mean (expressed as percentage), of the areal density measurements on the various types of varnished surfaces. The results are presented in Tables 4–6 for the various lava flow surfaces, the intact varnish areas surrounding the rock art, and the individual rock art elements themselves.

On the Aberdeen lava flows (Table 4), measurements were made on rock faces within an area of about 5–10 m across on each flow unit, choosing faces in all cardinal directions and a full range of inclinations from near horizontal to near vertical. The coefficients of variation of the measurements made on the same flow unit averaged 47% (37–62%) for $D_{Mn}$ and 68% (55–87%) for $D_{Fe}$. The higher CV for $D_{Fe}$ is mostly related to the need to subtract a relatively high and somewhat variable Fe background related to the underlying basalt (5.8±0.23 mass-%) from the readings on the varnish (Range 5.5–9.2 mass-%). Since 6 to 18 measurements were made on each flow, it follows that the uncertainty of the mean areal densities (i.e., the standard error) is about 10–20% at the spatial scale of sampling, i.e., some 2–10 m.

The data on the Red Hill and Little Lake basalts in Table 4 are from measurements made on intact varnish adjacent to petroglyphs and supplemental measurements made at a number of spots on the flows. They are spatially distributed over scales of a few tens of meters in the case of the flood-scoured area on the Red Hill basalt at Fossil falls, and up to a few km for the other spots on the Red Hill and Little Lake flows. The variability is comparable to the more closely spaced data from the Aberdeen flows, with the exception of the $D_{Fe}$ from the flood-scoured area. This unusually high scatter results from the inclusion of some near-zero measurements from a small area, which may be related to an unusually low Fe content in the underlying rock at this point. This explains in particular the extremely large error on $D_{Fe}$ on element FFS-2.

The measurements on the intact varnish adjacent to the petroglyphs (Table 5) probe smaller spatial scales (of the order of centimeters to about a meter). Consequently, the CVs of the

intact surfaces are smaller than those from the lava flows, averaging 21% (7–39%) for $D_{Mn}$ and 19% (2–53%) for $D_{Fe}$. Thus, when four measurements of intact varnish are made for a given rock art element, a statistical uncertainty of about 12% can be expected for the mean.

The variability of the measurements within the rock art elements (Table 6) is slightly larger than that in the surrounding varnish, averaging 28% (0–83%) for $D_{Mn}$ and 27% (3–145%) for $D_{Fe}$. This is likely due to the lower areal densities of the varnish on the petroglyphs and the resulting larger relative measurement error. Some of the variability could also result from residual small pockets of varnish in the pores of the basalt, which had not been removed by the artist when the petroglyphs were pecked. Care was taken during the measurements to avoid such spots with residual varnish, but sometimes very small spots may not have been visible. The normalized data have the same CVs as the absolute densities, since the measurements on each petroglyph were always normalized by dividing them by the average of the measurements on the corresponding intact varnish. Their statistical uncertainty can be estimated by error propagation from the CVs of the petroglyph and intact measurements and the corresponding number of replicates (columns S.E. in Table 5); they average 20% (5–54%) for $N_{Mn}$ and 18% (4–52%) for $N_{Fe}$.

As a qualitative footnote on extreme small-scale variability, we show an image of the varnish around a large olivine phenocryst on a near-vertical rock surface at Little Lake (S5 Fig). The olivine is weathering faster than the surrounding basalt, and thus forms a depression in the surface. In the area immediately around the olivine, no varnish is present, probably because the Fe(II) dissolving from the olivine creates reducing conditions and prevents the formation of Mn(IV). Around this whitish zone is a reddish halo, where Fe(III)-oxyhydroxides are present, gradating into the blackish, Mn-rich varnish that coats most of the rock. Just below the olivine is a metallic-black streak, indicative of a thick Mn-rich varnish, which may have formed from the Mn(II) dissolved during weathering of the olivine. This example may, on a macro-scale, represent the dissolution-oxidation-reprecipitation mechanism responsible for the varnish formation from deposited dust.

Another cautionary example of small-scale variability, driven by microbial activity and potentially affecting varnish areal density was provided to us during the review process by R. Dorn, and is presented in the (S1 Appendix).

## Absolute Mn and Fe accumulation rates

Measurements on surfaces of known age allow the determination of an effective or apparent element accumulation rate, $R_E$ (where E is Mn or Fe), calculated by dividing the areal density by the exposure age of the surface (Table 4). This rate is the average rate of element accumulation on the rock surface over the time it has been exposed, and as such averages over potential variations of the true, instantaneous accumulation rate with time. We use the term "apparent" to reflect this potential time-dependence and the possibility that in the long run, deposition is likely to compete with removal of varnish by erosion and weathering of either the varnish itself or of the underlying rock. If the instantaneous accumulation rate, $r_E$, were constant over time, $R_E$ would equal $r_E$, and a plot of Mn or Fe areal density against exposure time (age) should be a straight line going through the origin, since by definition there is no varnish on a newly exposed surface. Fig 3 clearly shows that such a linear model does not match our measurements, especially since any linear fit would intersect the y-axis far above the origin. The logarithmic model shown as a curve in Fig 3 provides a fairly good fit and can even go through the origin within its uncertainty, but is not well enough constrained at ages below 10 ka to be useful for dating purposes.

In this non-linear logarithmic model, the apparent Mn deposition rate, $R_{Mn}$, is a function of age, A. The instantaneous accumulation rate, $r_{Mn}(A)$, is the first derivative of the

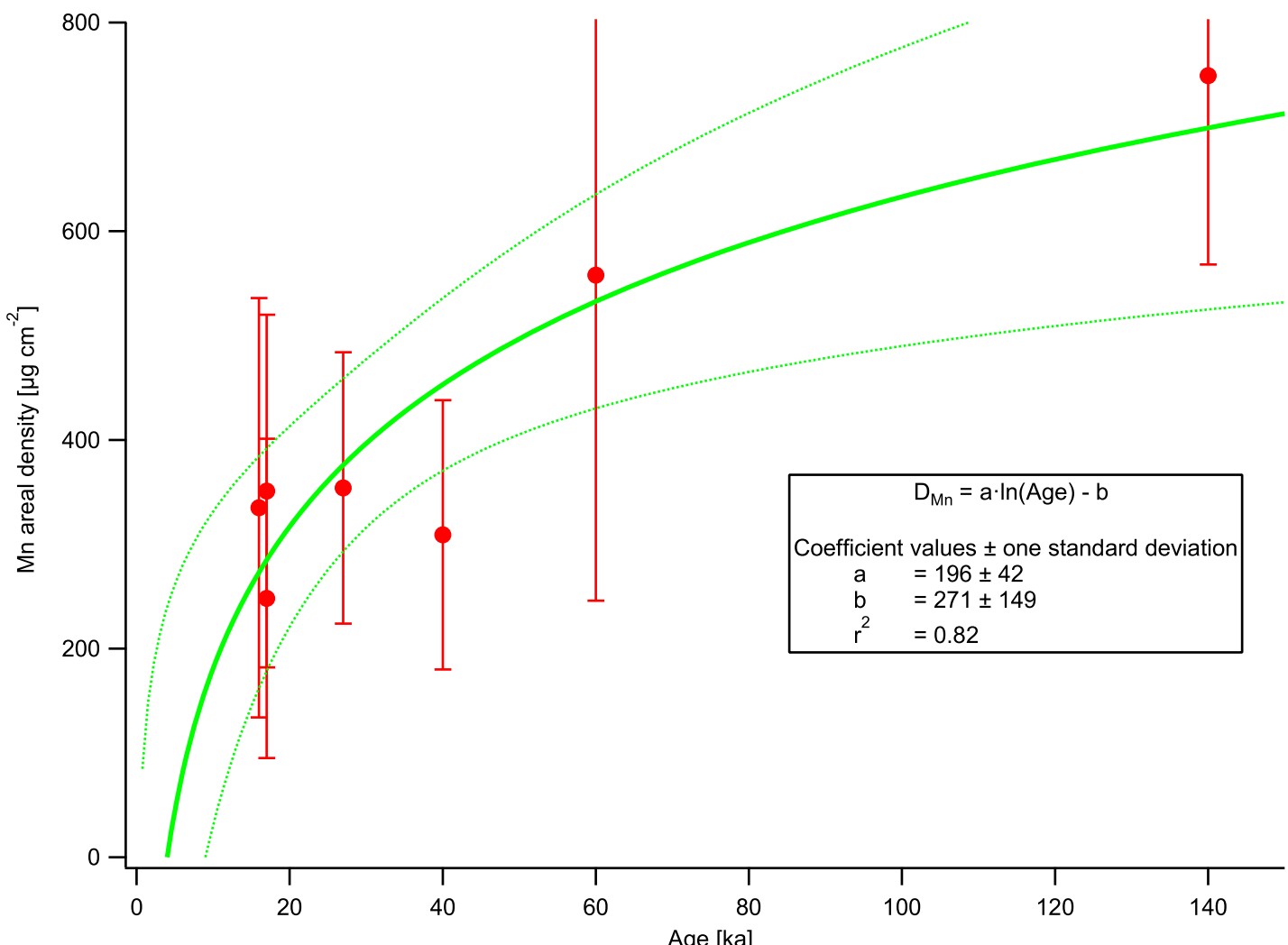

**Fig 3. Mn areal density versus surface age.** Plot of the Mn areal density, $D_{Mn}$, versus surface age, A, of rock varnishes on lava flow surfaces of known age. The error bars represent the standard deviation of the measurements on each lava flow surface. The solid line represents the fit equation, the dotted lines the 95% confidence interval of the fit.

logarithmic function $D_{Mn}(A)$ in Fig 3, i.e., a hyperbola, as is $R_{Mn}(A)$. Since ideally, $R_{Mn}(A)$ is the function that would be required for varnish dating, we show a plot of observed $R_{Mn}$ vs. A in a supplemental figure (S6 Fig). Clearly, $R_{Mn}(A)$ is not constant and can be fitted with a hyperbola, but this fit is not statistically robust (a constant $D_{Mn}(A)$ also yields a hyperbola) and poorly constrained below 10 ka, and is only shown to highlight its non-linear character.

While these results support the hypothesis that apparent varnish growth slows down with time and eventually comes to a standstill when growth is balanced by removal, they also imply that a growth rate obtained on older surfaces cannot be used to estimate the age of a much younger varnish deposit. In particular, given the steepness of the fits at young (<20 ka) ages, it is not legitimate to extrapolate the curve in S6 Fig to younger ages. Consequently, if Mn accumulation rates are to be used to estimate the age of Holocene varnished surfaces, e.g., petroglyphs, they would have to be calibrated using surfaces of comparable ages, such as Holocene lava flows.

On the other hand, the apparent Mn accumulation rates from the relatively youngest surfaces, i.e., flows CLS-04 (17 ka), CLS-05 (17 ka), and the flood-scoured surface at FF (16 ka),

group quite close together (20.6, 14.6, and 20.9 µg cm$^{-2}$ ka$^{-1}$, respectively) and are of the same magnitude as the averages of rates from mid- to late Holocene petroglyph surfaces from the Hima (13.4 µg cm$^{-2}$ ka$^{-1}$) and Ha'il regions (17 µg cm$^{-2}$ ka$^{-1}$) in Saudi Arabia [17, 42]. They are also comparable to the estimate of 30 µg cm$^{-2}$ ka$^{-1}$, which we derived from the measurements of Reneau [34] on Holocene surfaces in the Mojave Desert [17].

Using a technique similar to ours, McNeil [39] had found accumulation rates of 56–76 µg cm$^{-2}$ ka$^{-1}$ by pXRF measurements on 40–41 year old inscriptions on sandstone in Utah, about 3–4 times as large as our values on the 16–17 ka old basalt flow surfaces. While regional and substrate differences may play a role, this finding agrees with our previous observations of fast initial varnish growth [17, 42, 80]. It thus becomes evident, that a linear growth model is not applicable over extended periods of time, highlighting the need to find and measure dated surfaces spanning ages from decades to many millennia.

To compare our Mn accumulation rates with the thickness growth rates given by Liu and Broecker [19], we derive estimates using an average Mn concentration of 4.9% in the varnish (based on the ICPMS measurements on the Aberdeen flow varnishes) and a specific gravity of the varnish of 2.4 g cm$^{-3}$. This yields a range of 0.66 to 1.8 µm ka$^{-1}$, close to our values of 1.2–1.3 µm ka$^{-1}$ from Saudi Arabia and at the low end of the range of values (<1–40 µm ka$^{-1}$) in the compilation of Liu and Broecker [19]. It must be noted, however, that their measurements were made on the thickest spots in microbasins, typically on near-horizontal surfaces, thus representing the thickest varnish from a given site. In contrast, our growth rates represent areal averages over meters to kilometers, measured on inclined surfaces, and thus would be expected to be considerably lower.

## Rock varnish on petroglyph surfaces

One of the objectives of our study was to investigate to what extent pXRF measurements on the rock art could be used to assign relative or absolute dates to the rock art elements. Examples of petroglyphs showing typical rock art motifs from our sites are presented in Fig 4 and the complete set of analyzed elements is provided in S2 Fig. As outlined above, the study region has been occupied throughout the Holocene, and thus in principle ages between about 11,000 and zero years are possible. After some early attempts to establish rock art chronologies based on stylistic arguments [e.g., 24, 53], there have been several recent attempts to assign ages to the rock art in the Coso Range and at Little Lake, based on archaeometric, ethnographic, ecological, and archaeological evidence [e.g., 36, 51, 52, 55, 63]. While these studies differ in details, they agree on a number of basic points. Rock art production likely began around the Pleistocene/Holocene transition (ca. 11 ka BP), with abstract designs and some types of bighorn sheep being the earliest motifs [28, 36, 52]. At the other end of the time scale, rock art production appears to have continued well into the Numic period, possibly into the ethnographic period [36, 63]. Bighorn sheep representations were produced during the entire time, with the classic Coso sheep motif (Rogers' Type III) appearing around 4000 BP and becoming most frequent between about 2000 and 700 BP [52].

The atlatl motif may have appeared around 5000–7000 BP and mostly vanished around 1500 BP, after the atlatl was replaced by the bow and arrow around 2000 BP [52]. The patterned-body anthropomorphs (PBA) have been suggested by Rogers [52] to belong mostly in the Little Lake Period (ca. 7000–4000 BP), with another group of PBAs produced later (beginning around 3500 BP with atlatls and around 1050 BP with bows and arrows).

In the following discussion, we use the terminology given by Van Tilburg, Hull [51]. An "element" is a single form or design unit, often used synonymously with "petroglyph". A "motif" is an element that is often used within a given corpus and is related to a particular

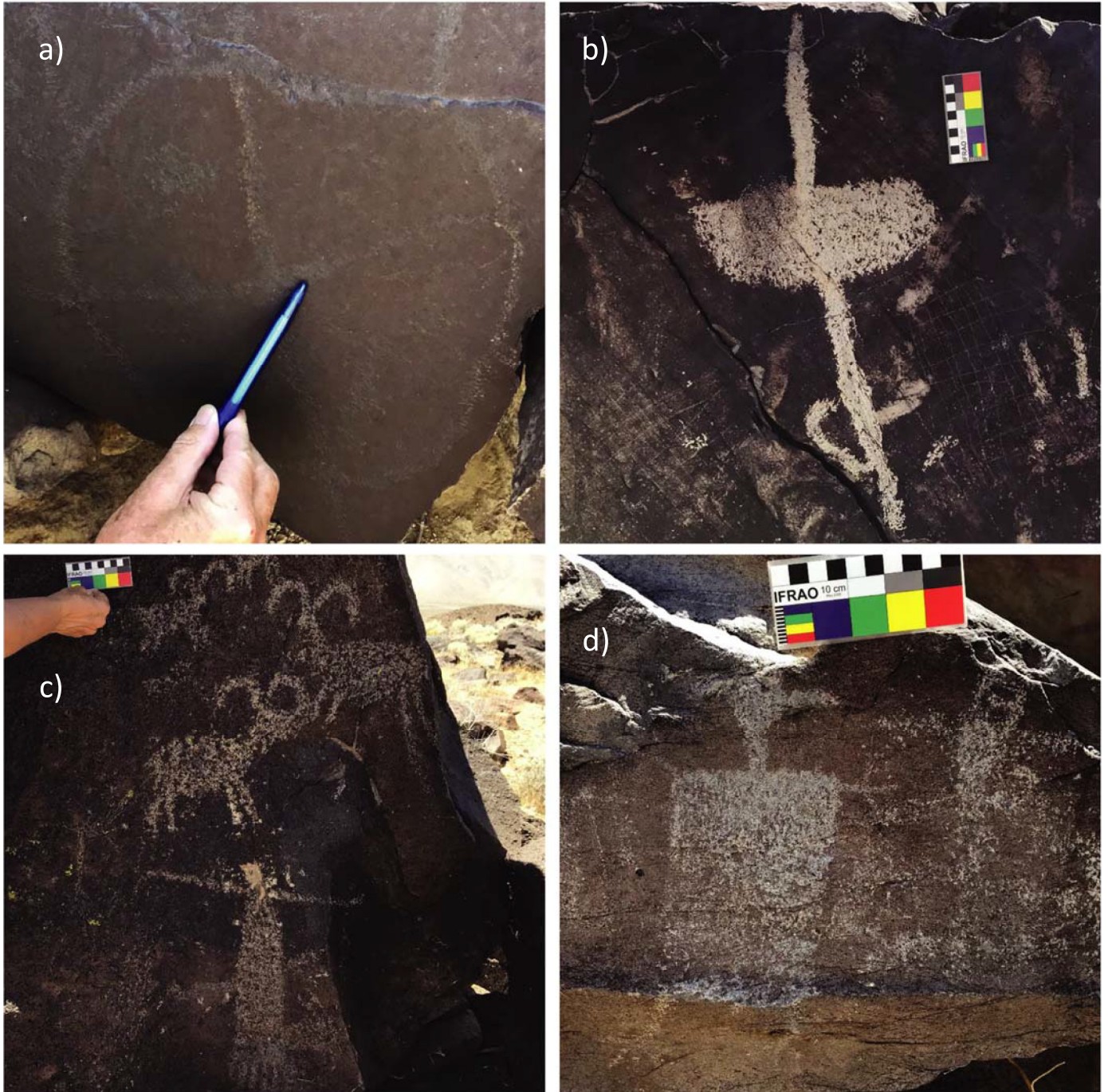

**Fig 4. Exemplary petroglyph images.** (a) Curvilinear abstract (LLA-1); (b) Atlatl (LLA-4); (c) Coso bighorn sheep and "shaman's bag" (FF-1, -2, and -3); (d) Anthropomorph (LL7-4). For photographs of all studied petroglyphs see the (S2 Fig).

style, e.g., an atlatl or a bighorn sheep. We also follow their practice of putting conventional terms for motifs, e.g., "shaman's bag" or "bear paw", in quotes on initial use, then without quotes.

Most of the petroglyphs in the study area have been produced by pecking; at Little Lake, a total of 4112 pecked rock art elements have been documented, in contrast to only 662 elements that had been created by scratching [51]. Because the scratched lines are narrower than the pXRF spot size, they have not been investigated in this study.

In the following, we examine whether our Mn density measurements provide a basis for deriving absolute or relative age estimates for the Owens/Rose Valley rock art. Conceptually, this approach is based on the fact that at the moment of its creation, the Mn density on the rock art is zero (assuming all varnish has been removed), and that over time the varnish will regrow to match the surrounding rock surface. Thus, $N_{Mn}$ is a quantitative, chemical analog to the commonly used visual or spectrophotometric method of estimating rock art ages. Visual estimates were used by Whitley [108] as rough indications of the age of various Great Basin rock art motifs. At Little Lake, Bretney [37] used a spectrophotometric approach to achieve a relative chronological ordering of rock art elements at Atlatl Cliff. Similar visual and spectro-photometric techniques have been applied elsewhere by a number of authors [32, 33, 109]. Lytle, Lytle [38] applied a pXRF technique similar to ours to petroglyphs in the Coso Range; unfortunately this work has not been fully published and important details are not available.

For the purpose of deriving absolute ages, it is necessary to have calibration surfaces of known age, similar to the age of the rock art, on which $D_{Mn}$ or $N_{Mn}$ can be measured to determine the absolute or normalized Mn accumulation rate. In Saudi Arabia, we had benefitted from the presence of inscriptions or specific motifs, for which approximate ages were known. Unfortunately, there are no inscriptions in our study area, and the ranges of independently estimated ages of the rock art motifs span too long periods of time to be useful for calibration. We thus first examined the potential of using the Mn accumulation rates measured on the Late Pleistocene lava flows to derive age estimates for the rock art. Above, we have discussed our findings that the apparent accumulation rate decreases with age, and that the known ages of the basalt flow surfaces we investigated were all substantially greater than those expected for the rock art. We thus checked what ages would be obtained if we used the highest measured Mn accumulation rates in our study area (ca. 21 µg cm$^{-2}$ ka$^{-1}$, on CLS04 and the flood-scoured surface at FF) to derive age estimates. This approach yields age values as high as 29 ka, clearly far in excess of the possible ages of the rock art in the study area. This highlights again the need of making calibration measurements on surfaces with known ages that are comparable to those of the petroglyphs of unknown age.

Given our inability to derive absolute age estimates, we examined whether our data are able to provide relative ages that are consistent with the chronologies discussed above. In Fig 5A we have plotted the $N_{Mn}$ values of the petroglyphs grouped by motifs, with the point marker styles indicating their locations. The yellow shading in the figure indicates the ages associated with these motifs in the chronologies discussed above. The "implied" age scale on the right was chosen by making the hypothetical assumption that full revarnishing occurs in 10 ka, corresponding to a linear revarnishing rate of 10% ka$^{-1}$, in analogy to the revarnishing rates of 10–12% ka$^{-1}$ we had found in Arabia. In the following we examine whether the observed $N_{Mn}$ are consistent with the ages implied under this hypothesis.

The highest $N_{Mn}$ value is found for LLA-1, a curvilinear abstract element from Atlatl Cliff (Fig 4A). This is consistent with a potential earliest age of about 11 ka for Great Basin rock art and the findings of Van Tilburg and Bretney [76] who consider Atlatl Cliff the oldest rock art locus at LL. The $N_{Mn}$ values of the atlatl elements at Atlatl Cliff and FF are consistent with the period of 6 to 1.5 ka BP for the production of this motif, with the exception of one element from Atlatl Cliff where visual inspection shows clear indications of repecking. All of the Type III (Classic Coso) bighorn sheep fall within their error bars in the implied age range for this motif (4000 to 700 BP). The low degree of varnishing of the Type I and II bighorn sheep at LL

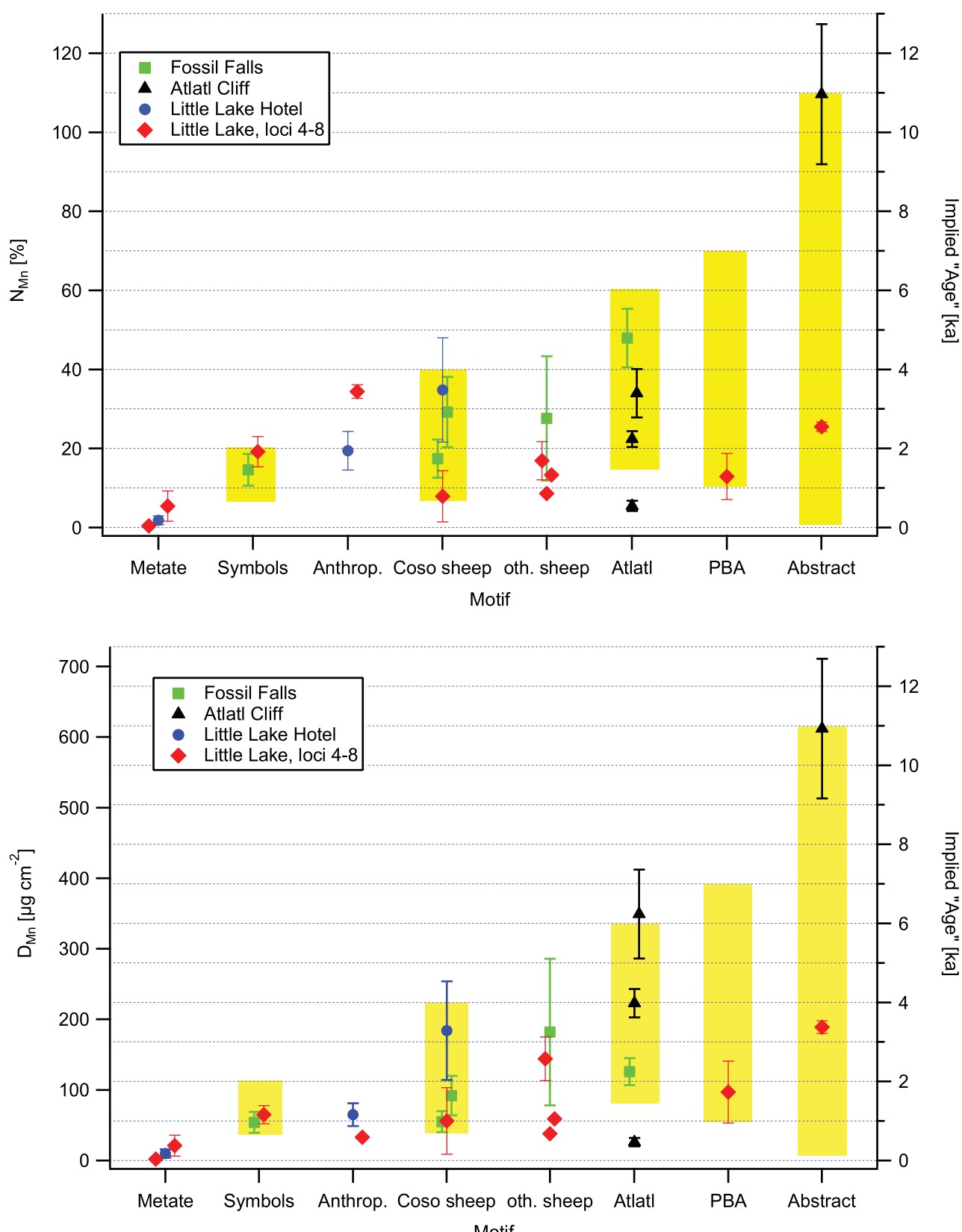

**Fig 5. Mn areal density on the rock art elements.** (a) $N_{Mn}$ values of the petroglyphs grouped by motifs; (b) $D_{Mn}$ values of the petroglyphs grouped by motifs. ("Symbols" includes "shaman's bag" and "bear paw" motifs; PBA: pattern-bodied anthropomorph). The yellow bars represent the ages of the motifs based on the chronology of Rogers (2010). Symbols: "Medicine bag" (green), "Bear paw" (red); PBA: patterned-body anthropomorphs; Anthrop.: other anthropomorphs. The error bars represent the standard deviations of replicate measurements on each rock art element.

(9–17%) is consistent with the florescence of bighorn sheep petroglyph production around 1300 BP proposed by Van Slyke and White [110], and the overall range of $N_{Mn}$ on bighorn sheep in the study region (8–35%, corresponding to implied ages of 800–3500 a) agrees with the focus on bighorn sheep hunting between 3500 and 800 BP suggested by Gilreath and Hildebrandt [55]. Van Tilburg and Bretney [76] associated the rock art production along the western side of the lake (Loci 4 and 8) and at Locus 7 with the latter part of the Newberry period (with a maximum between about 2200 BP to 1350 BP), and its continuation into the Haiwee and Marana periods, possibly extending until at least 1872 CE. Consistent with this finding, all but one of the $N_{Mn}$ measured at these loci (red diamonds in Fig 5A) are between 0.4 and 26%. This also applies to the patterned-bodied anthropomorph (PBA) from locus 7 (LL7-2), which shows a low degree of revarnishing with no visual sign of repecking, and thus probably was created near the end of the time range given by Rogers [52] for this motif (ca. 7000–1000 BP). The "shaman's bag" symbol from Fossil Falls (FF-3; green square in Fig 5A) fits well with the date range in the Rogers chronology, which places this motif in the Haiwee period (2000–700 BP). Overall, the range and distribution of implied ages from Fig 5A agrees with the VML dates given in Whitley and Dorn [36], who found an overall age range of 11,200 to <300 a, with the oldest age represented by an abstract motif and about half of their ages being 1500 a or less.

Consequently, our assumption of a revarnishing rate of about 10% ka$^{-1}$ provides estimates of implied ages roughly consistent with chronologies based on other techniques. In contrast, assuming a rate of twice this value (i.e., 20% ka$^{-1}$) would imply that almost all of the rock art elements fall into an age range of 400–2000 a, in clear disagreement with the published chronologies. Similarly, a value of 5% ka$^{-1}$ would imply unrealistically old ages for the rock art in this study.

In Fig 5B, we apply an analogous approach to $D_{Mn}$, hypothetically assigning a Mn accumulation rate, $R_{Mn}$, of 56 µg cm$^{-2}$ to obtain an implied "age" of 11 ka for LLA-1, the petroglyph with the highest $N_{Mn}$, and plotting the corresponding $D_{Mn}$ for the different elements. The overall result is qualitatively similar, albeit with a somewhat less satisfactory match with the published chronologies. This is consistent with the fact that the $N_{Mn}$ and $D_{Mn}$ in our data set are highly correlated, with an r$^2$ of 0.76, indicating relatively similar $R_{Mn}$ of the various surfaces. Notably, the $D_{Mn}$ on the abstract element LLA-1 (750 µg cm$^{-2}$) is the same as the average of the intact Little Lake basalt surfaces, which proves that its high $N_{Mn}$ is not an artefact of a low surrounding varnish density, and suggests a very old age for this petroglyph. Several other petroglyphs with visually similar degree of re-varnishing were observed at Atlatl Cliff, but for logistical reasons no measurements could be made on these surfaces. For images and further discussion on the intensely re-varnished elements at Atlatl Cliff, see Bretney [37].

While, in the absence of suitable calibration surfaces, our measurements yield only rough age estimates, they do allow some relevant conclusions. First, the grinding surfaces or metates all show very low $N_{Mn}$, implying that they have been used in relatively recent times. The $N_{Mn}$ of two of them (LLH-3 and LL8-1) are clearly greater than zero (1.8±0.9% and 5.4±2.8%, mean and standard error, S.E.), suggesting that this is not a result of contemporary vandalism, but possibly related to continued use by indigenous people in the last few centuries. Second, visual inspection of some petroglyphs, particularly the atlatl LLA-4 and the Type III bighorn LL8-3, shows signs of re-pecking. These elements have $N_{Mn}$ distinctly lower than the other atlatls and

Type III bighorns, and their implied ages are well below the range expected for these motifs. Note that the metate LL8-1 ($N_{Mn}$ = 5.4±2.8%) and the re-pecked bighorn LL8-3 ($N_{Mn}$ = 7.9 ±2.4%) are close to one another at the same locus and have statistically indistinguishable $N_{Mn}$, suggesting that use of the grinding surface and re-pecking of the petroglyph may be connected. Third, in agreement with previous authors [28, 36, 52, 75], our measurements indicate that rock art creation in Rose Valley continued over an extended period of time, possibly starting around the Pleistocene/Holocene transition, but certainly over several millennia and extending into the last few centuries.

## Summary and conclusion

We analyzed rock varnish from the Owens and Rose Valleys in the Mojave Desert of southern California by portable in-situ X-ray fluorescence on surfaces that range in age from the Late Pleistocene to the historic period. To complement these in-situ measurements, we collected varnish samples from the lava flows of the Big Pine volcanic field in Owens Valley near Aberdeen, California and analyzed them by fs-LA-ICPMS.

The rock varnishes had a composition characteristic of Type I varnish [7], with Mn, Fe, Si, and Al as the dominant elements, consistent with a mixture of Mn-Fe oxyhydroxides and clay minerals. The Mn/Fe ratios varied between about 0.4% and 2.5%, reflecting varnish growth predominantly under arid conditions. Higher Mn concentrations and Mn/Fe ratios tended to be present in older varnish, suggesting the presence of Mn-rich layers formed during wetter periods in the Pleistocene, whereas the petroglyphs had lower Mn/Fe ratios in agreement with varnish formation during the drier Holocene.

The varnish showed typical enrichments in a series of elements: Mn, Pb, Co, Ce, REY, Ba, Y, Zn, U, Th, V, and Fe. Among them, Pb, Ce, Cu, U, and Th were particularly strongly enriched, whereas the P and Ni enrichments were unusually low. In previous studies from other sites in the Mojave Desert and Death Valley, we have observed a similar pattern, suggesting a regional similarity of dust composition and enrichment processes. The REY enrichment pattern showed an unusually high Ce enrichment, a distinct negative Eu anomaly, and a slight negative Y anomaly, as well as an enrichment of the light REE and Y over the heavy REE. These enrichment patterns are consistent with a varnish formation process starting with the mobilization of Mn and trace elements from aeolian dust under mildly acidic conditions as they exist in atmospheric moisture (dew, rain), followed by increasing pH due to evaporation and mineral weathering reactions, which results in the abiotic or microbial oxidation of Mn, precipitation of Mn/Fe oxyhydroxides, and trace metal scavenging by the oxyhydroxides [12, 14, 16, 18, 95, 96, 111].

The areal densities of Mn in the rock varnish revealed an increase with age, from an average of 160±170 µg cm$^{-2}$ in the petroglyphs to 750±280 µg cm$^{-2}$ on the 140 ka Little Lake lava flow. The densities in the present study area were substantially higher than at our previous sites in Arabia, in spite of similar precipitation rates between the sites and higher dust fluxes in Arabia. This may be related to a greater weathering resistance of the basalt host rock in this region compared to the sandstone substrate in Arabia. Apparent Mn accumulation rates in the varnish were calculated from the measured areal densities and the known ages of lava flow surfaces. They showed a clear dependence on surface age, with the highest rates on the youngest surfaces. This indicates that the Mn accumulation is not linear, but decreases with age, as had been previously suggested by us [17] and others [38]. This implies that if Mn accumulation is to be used for age estimation of rock art, it is essential to have calibration surfaces with known ages in the range of the ages expected for the rock art.

The normalized Mn areal densities on the petroglyphs, i.e., the density on the petroglyph divided by that on adjacent intact varnish surfaces, range from near 0% to about 100%, and

show a distinct relationship with the known or inferred age of the surfaces. The highest $N_{Mn}$ were measured on curvilinear abstract elements, considered to be the oldest rock art at Little Lake based on archaeological considerations [51, 76], whereas the lowest $N_{Mn}$ values were on grinding surfaces (metates) that appeared to have been used recently. Given that no rigorous absolute ages could be determined for the rock art due to lack of suitable calibration surfaces, we examined whether relative ages could be estimated based on the $N_{Mn}$ and $D_{Mn}$ measurements. For this purpose, we made the hypothetical assumption that the oldest rock art had an age of about 10 ka based on previous archaeological and archaeometric studies [36, 52, 63, 76, 110]. Arranging the studied rock art motifs in the order of increasing $N_{Mn}$ and $D_{Mn}$ yielded a sequence consistent with previously proposed chronologies. Further, assuming approximately linear growth over the millennial timescales involved gave "implied ages" in rough agreement with those based on previous archaeological and archaeometric studies. We conclude that rock art creation in the Rose Valley area extended over a long time period, beginning around the Pleistocene/Holocene transition and continuing into the historical period.

In conclusion, while at this time absolute rock art dating by pXRF measurements remains elusive because of the lack of suitable calibration surfaces and uncertainties about the rate of varnish accumulation, our study does provide evidence for a potential to provide relative ages and some rough estimates of absolute ages. In view of the scarcity of alternative dating methods, especially of techniques that do not require destructive sampling, even such rough estimates are very useful. This technique also allows documenting the authenticity of ancient rock art and can provide evidence for or against recent reworking of petroglyphs. Future research should focus on acquiring measurements across a wide range of varnish on dated surfaces, especially from the Holocene period.

## Supporting information

**S1 Table. Factors other than time known to influence varnish growth on petroglyphs.** (PDF)

**S1 Fig. Overview map of the study area in Owens and Rose Valleys.** (Map services and data available from U.S. Geological Survey, National Geospatial Program.) (PDF)

**S2 Fig. Images of the petroglyphs measured in Rose valley.** (PDF)

**S3 Fig. fs-LA-ICPMS spot measurement profiles on rock varnish samples.** The x-axis represents successive laser shots on the same spot. Each laser shot corresponds to a depth increment of about 50–100 nm. (PDF)

**S4 Fig. Mn surface density versus surface inclination.** a) without inclination correction, b) with correction using the "attenuated cosine" correction equation (see text). (PDF)

**S5 Fig. Image of the varnish around a large olivine phenocryst on a near-vertical rock surface at Little Lake.** (PDF)

**S6 Fig. Mn apparent accumulation rate versus surface age.** Plot of the Mn apparent accumulation rate, $R_{Mn}$, versus surface age, A, of rock varnishes on lava flow surfaces of known age. The error bars represent the standard deviation of the measurements on each lava flow surface.

The solid line represents the fit equation, the dotted lines the 95% confidence interval of the fit.
(PDF)

**S1 Appendix. Influence of microcolonial fungi on rock varnish at the Conejo Mine site, California.**
(PDF)

## Acknowledgments

We thank the manager and owners of Little Lake Ranch for permission to make measurements on their property, and Tom Hnatiw for providing the photograph of the pattern-bodied anthropomorph at Little Lake. Ronald Dorn provided valuable comments on the manuscript during the review process, as well as the data and image that are included in S1 Appendix.

## Author Contributions

**Conceptualization:** Meinrat O. Andreae.

**Data curation:** Meinrat O. Andreae.

**Formal analysis:** Meinrat O. Andreae.

**Funding acquisition:** Meinrat O. Andreae.

**Investigation:** Meinrat O. Andreae, Tracey W. Andreae, Alan Garfinkel, Klaus Peter Jochum, Brigitte Stoll, Ulrike Weis.

**Methodology:** Meinrat O. Andreae, Klaus Peter Jochum.

**Project administration:** Meinrat O. Andreae.

**Resources:** Meinrat O. Andreae.

**Supervision:** Meinrat O. Andreae.

**Validation:** Meinrat O. Andreae, Tracey W. Andreae, Klaus Peter Jochum.

**Visualization:** Meinrat O. Andreae.

**Writing – original draft:** Meinrat O. Andreae.

**Writing – review & editing:** Meinrat O. Andreae, Abdullah Al-Amri, Tracey W. Andreae, Alan Garfinkel, Gerald Haug, Klaus Peter Jochum.

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
