## [Decision Letter · Decision Letter 0]

19 May 2020

PONE-D-20-09500

Geochemical studies on rock varnish and petroglyphs in the Owens and Rose Valleys, California

PLOS ONE

Dear Prof. Dr. Andreae,

Thank you for submitting your manuscript to PLOS ONE. After careful consideration, we feel that it has merit but does not fully meet PLOS ONE’s publication criteria as it currently stands. Therefore, we invite you to submit a revised version of the manuscript that addresses the points raised during the review process.

I agree with most of comments from the three reviewers and I would be happy if you will consider them while revising your manuscript.In recent years, you contributed to the development of rock varnish studies and this further study will represent a further step in this direction. A number of changes are required and I hope these will help to improve the clarity of the manuscript.

Besides the scientific comments, I would invite you to clarify about sampling strategy and to explain if you have the permissions from relevant US authorities (local? State? Federal?) to perform fieldwork, to collect samples of rock varnish from rock art galleries and to export them for lab analyses. This ethical statement is crucial and it is part of the editorial policy of PLoS ONE in the case manuscripts report on scientific investigation on cultural heritage.

We would appreciate receiving your revised manuscript by Jul 03 2020 11:59PM. To enhance the reproducibility of your results, we recommend that if applicable you deposit your laboratory protocols in protocols.io, where a protocol can be assigned its own identifier (DOI) such that it can be cited independently in the future. For instructions see: http://journals.plos.org/plosone/s/submission-guidelines#loc-laboratory-protocols

We look forward to receiving your revised manuscript.

Kind regards,

Andrea Zerboni, Ph.D.

Academic Editor

PLOS ONE

Journal Requirements:

3. We note that the Supporting Information Figures in your submission contain [map/satellite] images which may be copyrighted. All PLOS content is published under the Creative Commons Attribution License (CC BY 4.0), which means that the manuscript, images, and Supporting Information files will be freely available online, and any third party is permitted to access, download, copy, distribute, and use these materials in any way, even commercially, with proper attribution. For these reasons, we cannot publish previously copyrighted maps or satellite images created using proprietary data, such as Google software (Google Maps, Street View, and Earth). For more information, see our copyright guidelines: http://journals.plos.org/plosone/s/licenses-and-copyright.

1.    You may seek permission from the original copyright holder of the Supporting Information Figures to publish the content specifically under the CC BY 4.0 license.

Additional Editor Comments (if provided):

Reviewers' comments:

Reviewer's Responses to Questions

**Comments to the Author**

1. Is the manuscript technically sound, and do the data support the conclusions?

Reviewer #1: Yes

Reviewer #2: Yes

Reviewer #3: Partly

2. Has the statistical analysis been performed appropriately and rigorously? 

Reviewer #1: Yes

Reviewer #2: Yes

Reviewer #3: No

3. Have the authors made all data underlying the findings in their manuscript fully available?

Reviewer #1: Yes

Reviewer #2: Yes

Reviewer #3: Yes

4. Is the manuscript presented in an intelligible fashion and written in standard English?

Reviewer #1: Yes

Reviewer #2: Yes

Reviewer #3: Yes

5. Review Comments to the Author

Reviewer #1: Review comments: this paper studies rock varnish and petroglyphs in Owens and Rose Valleys, California by XRF and fs-LA-ICP-MS. Detailed geochemical features are given and the authors explained the elemental enrichment patterns. Based on previous archaeological studies, the authors tried to date the relative age of these rock art. The results are reliable and worthy of publication. I think this manuscript can be accepted after moderate revision. The paper should be modified in the two following aspects.

1. This content of this paper is too long and redundant. The language should be brief and to the point, especially for the introduction and methods part. Here I just list some examples that you need to revise. For line 96-120, you do not need to spare so much space to demonstrate what you have done in your previous work. For the geological background part, you do not need write much detail about local vegetation species, climate fluctuation history, human source and occupation history in this region, etc. Most of these information is irrelevant to your work. And please ONLY cite references which are useful to support you point of view.

2. I am not satisfied with the existing sequence of your figures. You first show the raw data, then the correlation curves and last the photograph of rock art. I think it is much better to show the satellite image of sampling site and photos of rock varnish first, and your laboratory analysis should be put last.

Reviewer #2: Rates of varnish accretion are highly dependent on microenvironment. Sometimes, the word microenvironment is misinterpreted to mean aspect or the direction an outcrop faces. Examination of south-facing versus north-facing aspects in a particularly dry environment could very well reveal no significant differences in accretion rates. In more mesic settings, however, aspect can play a major role. So can other forcing factors on microclimate such as cold-air drainage, subtle rainshadow effects, major rainshadow effects, inversions, the thermal conductivity of the host rock material, proximity to soil moisture such as along a wash that would have collected runoff. This small list only begins to account for variability in moisture availability on rock surfaces.

The impact of microenvironment has been noted in the only known method to measure both varnish thickness and varnish age, varnish microlaminations. The authors are very familiar with the papers of Tanzhou Liu and co-authors. Certainly Dr. Liu is the world’s foremost authority on the issue.

David Whitley’s seminar paper on the use of VML in the dating of petroglyphs in the southwest (Whitley, 2013) involved blind testing of Dr. Tanzhuo Liu and myself (Ron Dorn). We analyzed each other’s ultra-thin cross-sections and Dr. Whitley compiled the findings with his world-renowned expertise for rock art. The result was synthesized and condensed down. What was not apparent in the paper but was apparent to all of us (Dave, Tanzhuo and myself) was the highly variable thicknesses of varnishes with similar ages – and this is due to the effect of microclimate.

My own research on VML and its use certainly validates Dr. Liu’s findings. I used VML to estimate the ages of debris flows in metropolitan Phoenix (Dorn, 2010) and noted this problem, and yet the VML method was still able to predict a historical debris flow event that occurred four years later (Dorn, 2016). In other research project, variable rates of varnishing was also noted to be highly dependent on microenvironment (Dorn, 2014).

However, there are other papers taking different approaches to understanding varnishing that indicate the same thing: that microenvironment greatly influences varnish rates of accretion. The lead accumulation research of Spilde and colleagues was made possible because the more mesic varnishes they analyzed had rates of accretion faster than those found in more zeric settings (Spilde et al., 2013). A critique of the cation-ratio dating method notes a similar problem (Krinsley et al., 1990).

In an earlier paper on the use of XRF and Mn abundance to estimate petroglyph ages from Saudi Arabia, I recommended the authors put together a table of complications that could interfere with the use of their technique in estimating age. I was very pleased with the table they produced (Macholdt et al., 2019), and I suggest that they reproduce this table here in this paper. Some of the factors that they analyze could be in play, and the readers would benefit from their detailed analysis.

I also offer for the author’s use and give my permission for its use in this paper, an illustration that relates to the research presented by Whitley (2013). One of the petroglyphs analyzed in his Table 1 is CM15 an “X-motif”. The original publication of the cation-ratio age of 12000±600 yrs cal BP was not too dissimilar to the VML age confirmed by both myself and Tanzhul Liu of ca. 11,100 yrs cal BP — keeping in mind that both are minimum ages for the underlying engraving. What this one result in the table does not convey involved the reality that this petroglyph would not have shown much Mn-accumulation with the XRF approach.

Please examine the illustration that I attach in high resolution. One of several VML ultra-thin sections is shown on the right. To obtain these ultra-thin section was a real pain! Almost all of the tiny samples from the petroglyph were covered with microcolonial fungi that dissolve Mn and Fe and mobilize varnish. This means that the total mass of varnish in no way reflects its minimum age. The process to find VML sequences that were not disturbed by MCF was laborious and painstaking.

Note: the left images are secondary electrons (upper left) that show the microcolonial fungi, and back-scattered electrons (lower left) that show atomic number. Please note the areas with voids in the varnish that reflect another issue entirely … that even if the MCF do not dissolve the varnish completely, the organic acids (my assumption) can still leach Mn-Fe in variable amounts that are not time dependent.

Why did I highlight this particular petroglyph and these images? This petroglyph was in a location where water would accumulate in a sandy wash. The water from a tributary drained a region with relatively impermeable sediment. This meant that runoff would drain to the main wash near this petroglyph. The water would sink into the main wash, and humidity levels are much higher. The net result was an abundance of MCF growing on rock surfaces, including the sampled petroglyph.

This is but one example of a microenvironment that can cause several issues complicating the method that the authors use: (1) relatively fast rate of varnishing because of the relatively moist setting in a dry region; (2) complete removal of varnish by microcolonial fungi; (3) leaching of Mn-Fe from the varnish. The first complication works in opposition to #2 and #3, and there is no way to know what factor influences Mn-accumulation more and how to extract time from the Mn accumulation

Why was this petroglyph sampled if it would be such a pain? It was because Dr. Whitley thought it was particularly critical and asked that the effort be made to find a suitable sample.

Thus, you can see that after years of making thin sections and examining cross-sections, I am incredibly skeptical that any method of dating petroglyphs involving the accumulation of manganese can yield reliable results.

WHY DID I RECOMMEND ACCEPTANCE WITH MINOR REVISIONS IF I AM SO SKEPTICAL? The reason is that the authors did an outstanding job of presenting their research. They are clearly excellent scientists and I think the research needs to be published. The writing is excellent. The presentation of methods and results is clear. The literature analyzed is appropriate and balanced.

WHY DID I RECOMMEND MINOR REVISIONS? I would urge the authors to consider adding a table similar to the one they presented in Macholdt et al. (2019). I am sure that their thinking on potential complications with their method have evolved since they presented their outstanding table. The table could be modified to deal with the realities of

working with petroglyphs and study sites in the southwestern Great Basin of the USA.

I do not feel like it is my place to suggest that publication be conditional on the presentation of this cautionary table. I respect the authors and would trust their judgement given the perspective I am providing as a reviewer with substantial experience in this field area.

I also do not feel like it is my place even urge the authors to include the illustration I provide as an example of a petroglyph that would be highly problematic to analyze with their technique. Again, the authors fully understand the different approaches ... their approach goes for a much larger surface area. The one that I have found to be most

reliable and robust tries to find the best sampling area.

I appreciate the opportunity to provide the authors this feedback.

Dorn, R.I., 2010. Debris flows from small catchments of the Ma Ha Tuak Range, Metropolitan Phoenix, Arizona. Geomorphology 120, 339-352.

Dorn, R.I., 2014. Chronology of rock falls and slides in a desert mountain range: Case study from the Sonoran Desert in south-central Arizona. Geomorphology 223, 81-89.

Dorn, R.I., 2016. Identification of debris-flow hazards in warm deserts through analyzing past occurrences: Case study in South Mountain, Sonoran Desert, USA. Geomorphology 273, 269-279.

Krinsley, D., Dorn, R.I., Anderson, S., 1990. Factors that may interfere with the dating of rock varnish. Physical Geography 11, 97-119.

Macholdt, D.S., Jochum, K.P., Al-Amri, A., Andreae, M.O., 2019. Rock varnish on petroglyphs from the Hima region, southwestern Saudi Arabia: Chemical composition, growth rates, and tentative ages. The Holocene 29, 1377-1396 DOI: 10.1177/0959683619846979.

Spilde, M.N., Melim, L.A., Northrup, D.E., Boston, P.J., 2013. Anthropogenic lead as a tracer for rock varnish growth: implications for rates of formation. Geology 41, 263-266.

Whitley, D.S., 2013. Rock art dating and the peopling of the Americas. Journal of Archaeology 2013 http://dx.doi.org/10.1155/2013/713159, 1-15.

Reviewer #3: Meinart et al., set out to test whether they can date petroglyphs in Owens and Rose valleys in the western US based on the re-varnishing rates of the petroglyphs. They produced a calibration curve based on pXRD and IC-PMS measurements of varnish formed on basalt of known ages. In the end, they don’t use the calibration curve from the western US to derive the re-varnishing rate but one from Saudi Arabia that they previously calculated.

Firstly, I applaud the work Prof. Meinart and his students have been doing in the field of rock varnish research. However, there are a few issues in this work that must be addressed before this paper can be published.

Major

1. It took me a while to understand what exactly was done because of the terms “areal density” and “deposition rate”, which are a bit confusing. I suggest using the term “Lateral varnish growth rate”. In addition, it’s not deposition rate (1/depth or 1/m) but lateral coverage rate (1/area or 1/m^2) – how fast does varnish cover the rocks’ surface.

2. The question you are asking is how fast does varnish spread laterally. The most straight forward way to do this is by light image processing (as was previously done by Bednarik). The choice to use a pXRF complicates the ability to calculate the parameter you are looking for, because [Mn] is a function of two things: Is there varnish and what is the [Mn] concentration of the varnish (this in itself a function of vertical sedimentation rate and the climatic condition that the varnish formed in). In addition, as varnish is a sedimentary accretion, it first grows in micro-basins and once these are full it starts spreading laterally, so lateral varnish coverage should also be a function of surface roughness. So you have three unknowns and only one measurement, which makes this an underdetermined problem.

To try and illustrate my problem, it’s like using the lateral spread of quaternary sediments on a landscape to say something about the age of the sediments.

3. Your dating is based on the existence of a correlation of [Mn]/m^2/age vs. age (Fig. 4, lines 588-618). I’m not sure why you are dividing the lateral accumulation by the age and then plotting against age, this by construction would give you an exponential fit every time, with an r^2 value greater than 0.5. For example, I made a simple plot in excel, where column A is age and B is random numbers between (0-20) divided by age. You can see that you can get very high correlations generated in a random way (I chose a high r^2 value case). Because the age and 1/age have a perfect correlation with each other, and you divide this by a random number, on average you should get an r^2 value of at least 0.5. If the original data is slightly correlated with age (say r^2 =0.25) the combined r^2 value would be 0.75. This still means that the data and age are only correlated to 0.25, not 0.75..

This is why Liu and Broecker didn’t get a correlation and when you divided by age you did get one.

I suggest, plotting the lateral deposition (ng/m^2) against age, this is the metric you are interested in. How much varnish formed vs. time. Not how much varnish formed per time vs. time.

4. sampling strategy. For this method to be valid, you had to sample the basalt in a random way (otherwise your results are biased, say if you chose varnished areas over none varnished areas). From the methods it is unclear if this is the case. An example would be to draw a 3m line on the surface and measure every 10cm. Was a random method used? This needs to be clarified.

5. Do the concentrations of the major elements from pXRF correlate with major elements from the ICP-MS? Could this be used to further the ability to understand some of the scatter in the data?

6. The petroglyph age assessment. After the Owens data was rejected, the authors turn to use the Saudi data. But this data is 3-4 times slower than what is found in Owens (Lines 405-407). In addition, the age calculation used is linear (10% per 1ka). But the data (Figure 4) shows an exponential growth rate. Given the detailed description of the data, I’m not sure how the Saudi data can be used without a much more elaborate discussion.

Minor

Line 69 and 94. This processes is well described in Liu and Dorn (1996) – I recommend citing it.

Line 218. Please add how deep does the pXRF penetrate (if at all)? How does it deal with rough surfaces?

Line 229. Why are the measurements on the basalt valid and the petroglyphs need to be corrected?

Line 411. You use a very different method than Broecker and Liu’s 2001 work. They use very carefully chosen varnishes that display distinct features and then measure the average [Mn] of the downward profile of the varnish and compare that to rainfall amount. You are measuring the surface varnish in an 8mm diameter. I’m not saying Broecker and Liu are right, just that I think the difference in what you are measuring makes this statement problematic.

Line 449-458. Would differences in the amount of Fe in the dust make a difference on the Mn/Fe in the varnish?

Line 499. Please move this sentence to the methods and elaborate on what exactly was done. How many spot measurements are used to characterize one surface?

Line 531-535. These lines must be moved to the methods.

Line 550: How did you chose the surfaces adjacent to the petroglyphs for the comparison analysis? Are they of the same roughness?

Line 553-554: When measuring on a small scale you are improving the precision, but your accuracy might be off, i.e. it’s unclear if these few centimeters are representative of the larger population.

Fig 3. What do the error bars represent?

Line 612. remove comma after surfaces

Line 702. Fig. 4. What are the vertical error bars? In light of the discussion of lines 676-701 and the assignment of 33% uncertainty for the Saudi Arabia data (line 110) they seem very small to me. How did you aggregate the error?

Liu, T., Dorn, R.I., 1996. Understanding the Spatial Variability of Environmental Change in Drylands with Rock Varnish Microlaminations. Ann. Assoc. Am. Geogr. 86, 187–212. https://doi.org/10.1111/j.1467-8306.1996.tb01750.x

6. PLOS authors have the option to publish the peer review history of their article (what does this mean?). If published, this will include your full peer review and any attached files.

Reviewer #1: No

Reviewer #2: No

Reviewer #3: No

---

## [Editor Report · Decision Letter 1]

16 Jun 2020

Geochemical studies on rock varnish and petroglyphs in the Owens and Rose Valleys, California

PONE-D-20-09500R1

Dear Dr. Andreae,

We’re pleased to inform you that your manuscript has been judged scientifically suitable for publication and will be formally accepted for publication once it meets all outstanding technical requirements.

Kind regards,

Andrea Zerboni, Ph.D.

Academic Editor

PLOS ONE
---

## [Editor Report · Acceptance letter]

18 Jun 2020

PONE-D-20-09500R1 

Geochemical studies on rock varnish and petroglyphs in the Owens and Rose Valleys, California 

Dear Dr. Andreae:

I'm pleased to inform you that your manuscript has been deemed suitable for publication in PLOS ONE. Congratulations! Your manuscript is now with our production department. 

Kind regards, 

on behalf of

Prof. Andrea Zerboni 

Academic Editor

PLOS ONE